# Global determinants of freshwater and marine fish genetic diversity

Stéphanie Manel[1]*, Pierre-Edouard Guerin[1], David Mouillot[2,3], Simon Blanchet[4], Laure Velez[2], Camille Albouy[5] & Loïc Pellissier[6,7]

Genetic diversity is estimated to be declining faster than species diversity under escalating threats, but its spatial distribution remains poorly documented at the global scale. Theory predicts that similar processes should foster congruent spatial patterns of genetic and species diversity, but empirical studies are scarce. Using a mined database of 50,588 georeferenced mitochondrial DNA barcode sequences (COI) for 3,815 marine and 1,611 freshwater fish species respectively, we examined the correlation between genetic diversity and species diversity and their global distributions in relation to climate and geography. Genetic diversity showed a clear spatial organisation, but a weak association with species diversity for both marine and freshwater species. We found a predominantly positive relationship between genetic diversity and sea surface temperature for marine species. Genetic diversity of freshwater species varied primarily across the regional basins and was negatively correlated with average river slope. The detection of genetic diversity patterns suggests that conservation measures should consider mismatching spatial signals across multiple facets of biodiversity.

[1] CEFE, Univ. Montpellier, CNRS, EPHE-PSL University, IRD, Univ Paul Valéry Montpellier 3, Montpellier, France. [2] MARBEC, Univ Montpellier, CNRS, IFREMER, IRD, Montpellier, France. [3] Australian Research Council Centre of Excellence for Coral Reef Studies, James Cook University, Townsville, QLD 4811, Australia. [4] Centre National de la Recherche Scientifique (CNRS), Université Paul Sabatier (UPS); Station d'Ecologie Théorique et Expérimentale, UMR 5321, F-09200 Moulis, France. [5] IFREMER, unité Ecologie et Modèle pour l'Halieutique, Nantes, France. [6] Swiss Federal Research Institute WSL, CH-8903 Birmensdorf, Switzerland. [7] Landscape Ecology, Institute of Terrestrial Ecosystems, Department of Environmental System Science, ETH Zürich, CH-8092 Zürich, Switzerland. *email: stephanie.manel@ephe.psl.eu

The ongoing sixth mass extinction crisis, under ever increasing human pressures, urgently calls for a better understanding of the main processes shaping the distribution of biological diversity on Earth. However, most global studies are investigating biodiversity at the species level[1–4], while a few studies examine the diversity of genes within organisms, i.e. genetic diversity[5]. Indeed, the cost of sampling and genotyping a sufficient number of individuals within species has limited our understanding of the determinants of intraspecific genetic diversity, particularly at large scale. Spatial patterns of genetic diversity are mainly documented locally or regionally, mostly for a single species or a few species in phylogeographic[6] or landscape genetics studies[7,8].

Yet, determining the global distribution of intraspecific genetic variation and its main drivers is urgent, given that genetic diversity might be undergoing silent and poorly documented erosion under global changes[9]. Genetically distinct local populations may go extinct before the whole species does[10–12], resulting in the erosion of genetic diversity and adaptive potential for many species[13]. In this context, investigating the key determinants of genetic diversity patterns and their underlying biological processes would help to design comprehensive conservation schemes, i.e. protected areas, for this neglected component of biodiversity[14–16]. Surprisingly, there is currently only a limited description and comprehension of the large-scale organisation of genetic diversity[17,18].

Intraspecific genetic diversity might show biogeographic patterns congruent with those of species diversity as a result of processes acting along a micro- to macroevolution continuum[19–22]. Among the hypotheses explaining spatial congruence between intra- and inter-specific levels of diversity, the evolutionary speed hypothesis posits that higher temperatures foster higher metabolic and mutation rates, as well as faster generation times, which should in turn increase genetic divergence, speciation rate and, ultimately, species diversity[23]. Under this hypothesis, species and genetic diversity are both expected to be higher in warmer regions. A positive association between species, genetic diversity, and temperature is also expected under the "colonisation hypothesis" (or "stability hypothesis") where demographic fluctuations are associated with environmental instability which in turn limits diversity. These events are generally followed by stochastic recolonisation generating bottlenecks, which may lower both species and genetic local diversity[24]. Typically, warmer areas in the tropics have experienced less historical variability, whereas cold areas were highly unstable, generating species diversity clines along temperature gradients[4]. The energy hypothesis assumes that more productive areas sustain larger population sizes, which should favour higher genetic diversity and allow the persistence of more species along with, eventually, a higher speciation rate[25,26]. Finally, the physical complexity hypothesis states that areas with higher habitat complexity should provide more ecological niches and hence support higher species diversity[27,28], but also more spatially structured populations in a given area so a higher genetic diversity[29]. This physical complexity hypothesis strongly depends upon the spatial grain and extent of the study area. For instance, at a large scale, a complex network of watercourses, typically characterising freshwater habitats, should promote higher genetic (and species) diversity than more homogenous and continuous marine waters.

Miraldo et al.[18] were the first to take advantage of the vast and ongoing accumulation of georeferenced genetic information on DNA sequences. From a compilation of thousands of short genetic sequences for terrestrial vertebrates (<600 bp), they revealed higher genetic diversity in tropical than in temperate regions. Although this finding seems coherent with known patterns of species diversity in vertebrates[30], the extent to which intraspecific genetic and species diversity show similar distributions across regions or ecosystems remains to be explored[31].

Ray-finned fishes (Actinopteyigii) are an old clade of vertebrates that radiated into diverse habitats including marine and freshwater environments, from the tropics to the poles[32,33]. They represent a fascinating case study to investigate the association between genetic diversity, species diversity and the environment in different regions and ecosystems. Freshwater fish diversity (total number of species) is higher than marine fish diversity (~15,200 and ~ 14,800 species, respectively) while marine environments cover ~70% of Earth and 97% of all waters[34,35]. In marine ecosystems, fish species diversity is concentrated in coastal waters (depths of <200 m) that represent <1% of the world's sea surface[36]. Marine and freshwater fish diversity also declines with decreasing temperature at large spatial scale[3,37]. These global patterns suggest that differences in ecosystem productivity, environmental conditions and habitat connectivity or complexity likely shape fish species diversity[27]. Whether these patterns hold at the intraspecific level, i.e. genetic diversity, has not yet been investigated.

Here, from a macro-genetic perspective, we study the global distribution of genetic diversity in ray-finned fishes using data for 1611 freshwater and 3815 marine species. Genetic diversity patterns are produced by assembling 50,588 georeferenced mitochondrial sequences in the Barcode of Life Database (BOLD). We then estimate nucleotide diversity for each species in each grid cell at a spatial resolution of 200 km. This nucleotide diversity is averaged across the species in each cell. We first investigate the correlation between genetic and species diversity separately for marine and freshwater fishes. Next, we explore the global environmental and geographic determinants of the mean nucleotide diversity across species per cell, hereafter called genetic diversity. We interpret our results according to the micro-macro continuum concept and in the light of the evolutionary speed, colonisation, energy and habitat complexity hypotheses.

## Results

**Global patterns of fish genetic diversity.** In total 34,782 and 15,806 sequences were retrieved for marine and freshwater fish species, respectively, and were used to estimate the mean nucleotide diversity across species within each cell on a worldwide grid with a 200-km spatial resolution (Supplementary Tables 1 and 2). We showed that intraspecific genetic diversity in marine and freshwater species was heterogeneously distributed across the globe (Fig. 1) with a strong and significant signal of spatial autocorrelation (Supplementary Fig. 1). Regions with high genetic diversity (above the 90th percentile; top 10% richest cells) are located in the Western Pacific, the North Indian Ocean and the Caribbean seas for marine species (Fig. 1a; Supplementary Fig. 2a), and in South America for freshwater species (Fig. 1c, Supplementary Fig. 2c). Regions with low genetic diversity (below the 10th percentile of the distribution) are located in the North-Eastern and Western Atlantic and the Southern Atlantic for marine species (Fig. 1a, Supplementary Fig. 2b) and in Europe, Asia and North of South America for freshwater species (Fig. 1c, Supplementary Fig. 2d). When averaging genetic diversity across cells within latitudinal bands of 10°, a peak is observed at latitude 10°–20° S for both marine and freshwater species (mode = 0.025 and 0.036, respectively, Fig. 1b, d).

**Congruence between fish genetic and species diversity patterns.** We found a positive and significant, albeit weak, relationship between genetic and species diversity for both marine and freshwater fishes (Fig. 2; modified $t$-test for spatially dependent variables = 0.21; $p = 0.010$ for marine species;

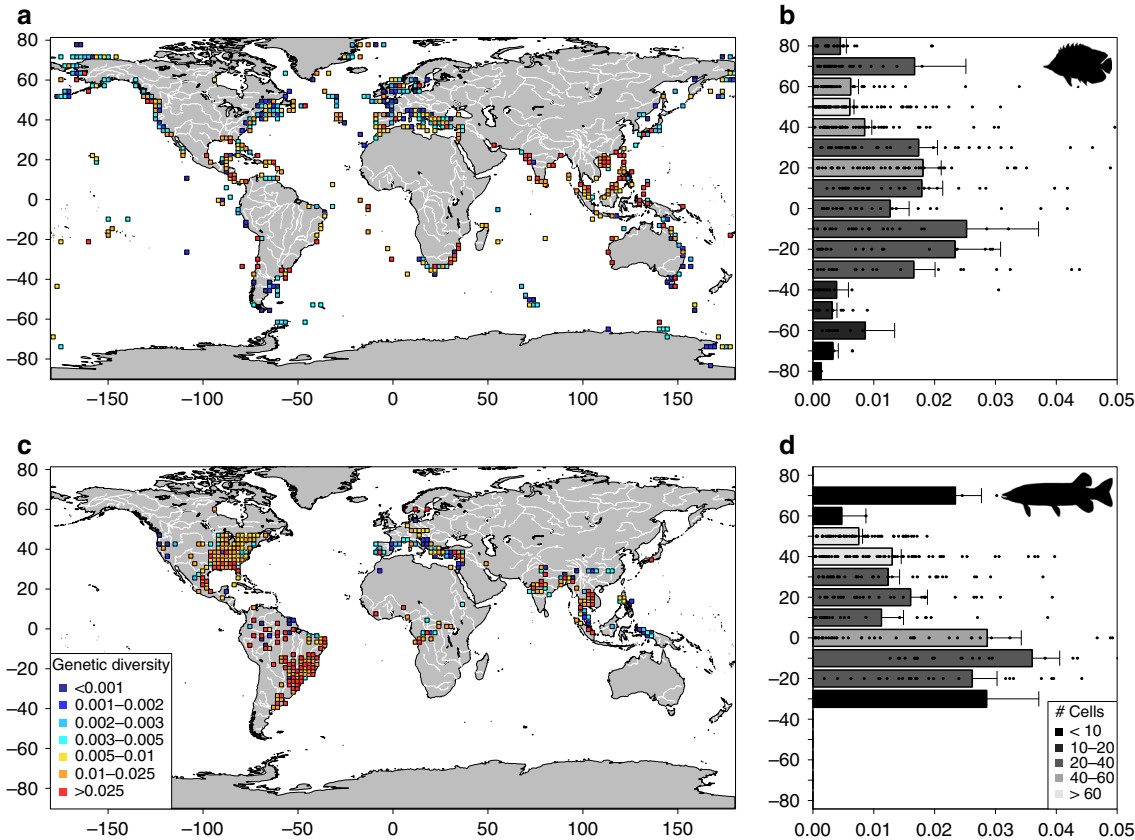

**Fig. 1 Biogeographic patterns of fish genetic diversity.** Genetic diversity was estimated as the mean number of mutations per base pair for Cytochrome Oxidase Subunit 1 sequence across species (**a**) 514 cells for marine fishes and (**c**) 343 cells for freshwater fishes. The colour gradient represents the relative variation of intraspecific genetic diversity: the reddest square cells have the highest genetic diversity. The colour scale of Fig. 1a, c is defined in the Fig. 1c: the bluest square cells have the lowest genetic diversity. Genetic diversity was averaged across cells within latitudinal band of 10° and is plotted as a function of latitude for marine species (**b**) and freshwater species (**d**) with error bars representing confidence intervals (standard deviation of mean genetic diversity across cells/square root of the number of cells) and indicates variability of genetic diversity among cells. The grey colour gradient indicates the number of cells used in each latitudinal band. The grey colour scale is defined in Fig. 1d. For the fish silhouette in Fig. 1d, credit given to U.S. Fish and Wildlife Service (illustration) and T. J. Bartley with licence at: http://www.phylopic.org/image/f8369dec-bdf6-432b-a0c4-41ee5d75286d/. Drawn with R version 3.2.3.

modified $t$-test = 0.36, $p = 0.015$ for freshwater species). Specifically, the median value of genetic diversity per cell is two times higher in freshwater (0.011; interquartile range: 0.0041–0.0200) than in marine fishes (0.0052; interquartile range: 0.0023–0.012), this difference being significant when accounting for latitude (Supplementary Table 3). Species diversity per cell tends to be higher for freshwater (median = 300 species; interquartile range = 109–741 species) than for marine fishes (median = 268 species; interquartile range = 97–797 species, Supplementary Fig. 3a), although the difference is not significant accounting for latitude (Supplementary Table 3). For freshwater fish, species diversity peaks in the South latitudinal band ranging 10°–20° as for genetic diversity, while for marine species the main peak is in the North latitudinal band of 30°–40° (Supplementary Fig. S3b, c).

**Relationship with environmental and geographic factors.** We used linear models to explore the relationship between fish genetic diversity per cell and three types of factors (environmental, geographic and sampling) (see Methods for more details, Supplementary Table 4). Since genetic diversity is spatially autocorrelated (Supplementary Fig. 1), we included an autocovariate to account for the spatial structure in our data that was not explained by our factors. This term integrates the spatial dependency among the 200-km spaced cells. The variance

inflation factor (vif), which assesses potential collinearity among factors (here we chose to remove all factors with a vif > 5), eliminated oxygen concentration for marine species. Oxygen concentration was highly and negatively correlated with sea surface temperature ($r = -0.98$, $p < 0.001$). For freshwater, river slope range was removed from the analysis because it was identified as highly collinear with average river slope (vif > 5).

The most parsimonious models were selected based on the lowest Akaike Information Criterion (AIC) to obtain parameter coefficients and partial plots. For marine species, sea surface temperature, regions, the spatial autocovariate and the number of species were retained in the final model (Supplementary Table 5). For freshwater species, air temperature, average slope, regions and the spatial autocovariate were retained (Supplementary Table 5) while elevation, basin area and flow accumulation were not selected in the final model. There was no residual autocorrelation signal in the final models (Moran $I = -0.05$, $p = 0.76$ for marine species and Moran $I = -0.018$, $p = 0.573$ for freshwater species).

The model for marine fish explained 16% of the variation in genetic diversity globally (Supplementary Table 6). Genetic diversity increased positively with sea surface temperature (Fig. 3a), which had the highest level of explanation (relative variance: 75%; Fig. 3b). The region factor indicated slightly higher genetic diversity in the Indo-Pacific than in the Atlantic region (Fig. 3a; Supplementary Fig. 4a). For freshwater fish, the most

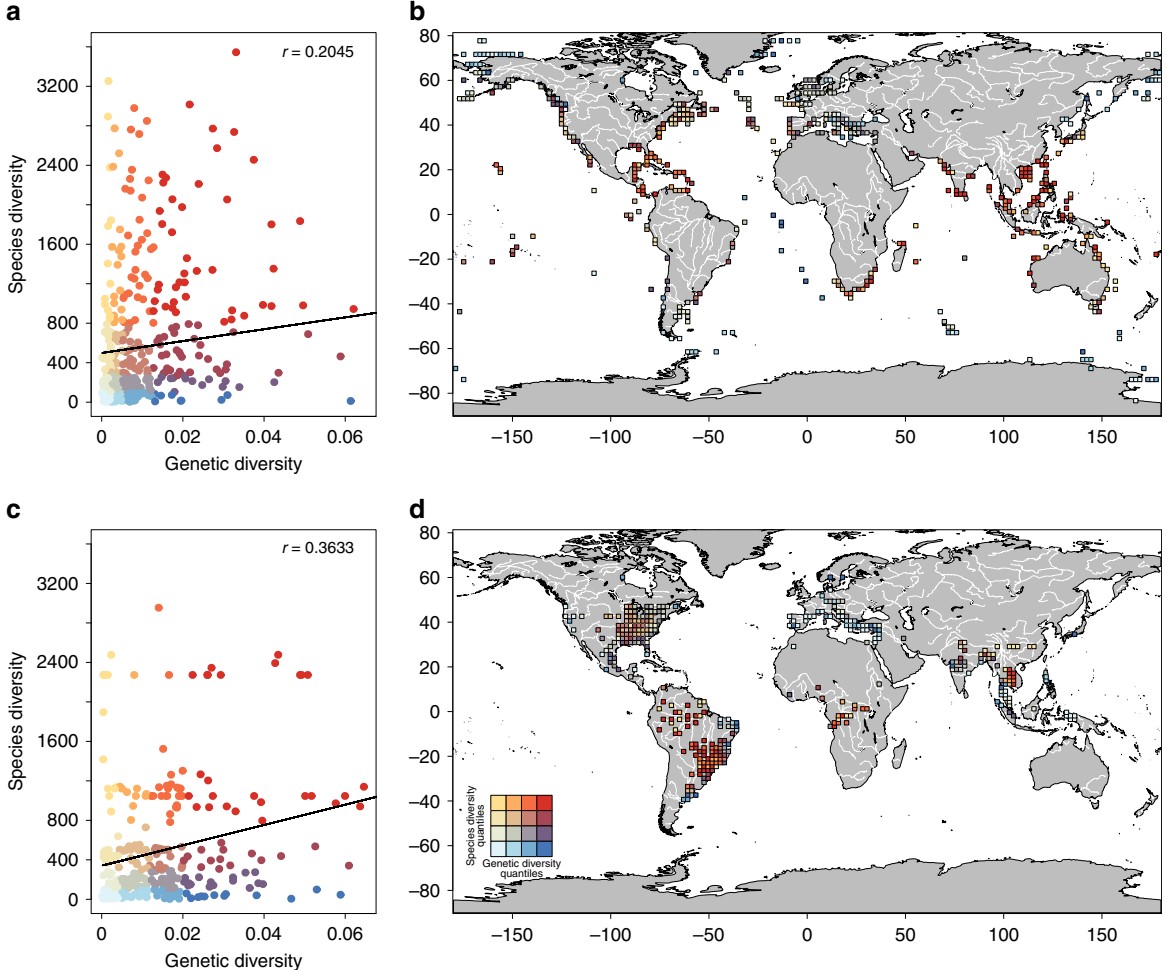

**Fig. 2 Congruence between fish genetic and species diversity.** Classification of cells depending simultaneously of their values of genetic and species diversity for marine (**a**) and freshwater (**c**) species. Values of diversity were reported on the global map using a colour gradient depending on the values of the genetic and species diversities for marine species (**b**) and freshwater species (**d**) respectively. The colour scale of Fig. 2a–d is defined in the Fig. 2d. The line was represented as the output of a linear model (lm) of the correlation between genetic diversity and species diversity. Person coefficient of correlations calculated in linear regressions (r) are reported on the figure. Drawn with R version 3.2.3.

parsimonious model explained 19% of the variation in genetic diversity globally (Supplementary Table 6). South America hosted fish populations with significantly higher genetic diversity than the other regions (Wilcoxon test = 18142, $p < 0.0001$) (Fig. 3c; Supplementary Fig. 4b). The geographical factors, including regions, average slope and the spatial autocorrelation factor explained the highest cumulative relative variance (88.9%) compared to the other factors (Fig. 3d).

In order to test the influence of the number of sequences and species in each cell as well as cell taxonomic coverage on the estimate of genetic diversity and model outputs, we re-ran both the modified $t$-test for spatially dependent variables applied on genetic and species diversity, and the models while selecting cells with more stringent filters (≥5 sequences, ≥8 species and ≥5% of taxonomic coverage) (Supplementary Tables 7 and 8). Filtering for taxonomic coverage per cell when estimating genetic diversity decreased the correlation between genetic and species diversity in comparison to the values obtained in the main analysis (≥2 sequences; ≥2 species; no taxonomic coverage) (Supplementary Table 7). The modified $t$-test of the spatial dependence was not significant when the taxonomic coverage per cell was higher than 1% for marine species and 0% for freshwater species (Supplementary Table 6). Conversely, filtering the number of sequences or species used as surrogates for the sampling effect

always increased the values of the modified $t$-test, except for marine species with more than three sequences, which were almost always significant (Supplementary Table 6). For marine species, removing cells with less than three sequences (at least 2 species), or filtering for a taxonomic coverage >2% when estimating genetic diversity decreased the effect of temperature (Supplementary Table 8). However, the explanatory power (adjusted $r^2$) of the models always increased with more stringent filters except when only <34% of cells were retained and filtering for cells with at least two species and with more than four sequences or with a taxonomic coverage per cell of 5% in each cell. For freshwater species, the effect of average river slope always decreased with more stringent filters (lower absolute values). The effect is less clear on the region coefficient but this main factor (region) was significant in all cases (Supplementary Table 8). The explanatory power of the model (adjusted $r^2$) increased in all cases.

## Discussion
Here we show that the genetic diversity of marine and freshwater fishes is not distributed uniformly across the globe but displays clear biogeographic patterns (Figs. 1 and 2). The congruence between genetic and species diversity is weak but significant, for both marine and freshwater fishes, suggesting common

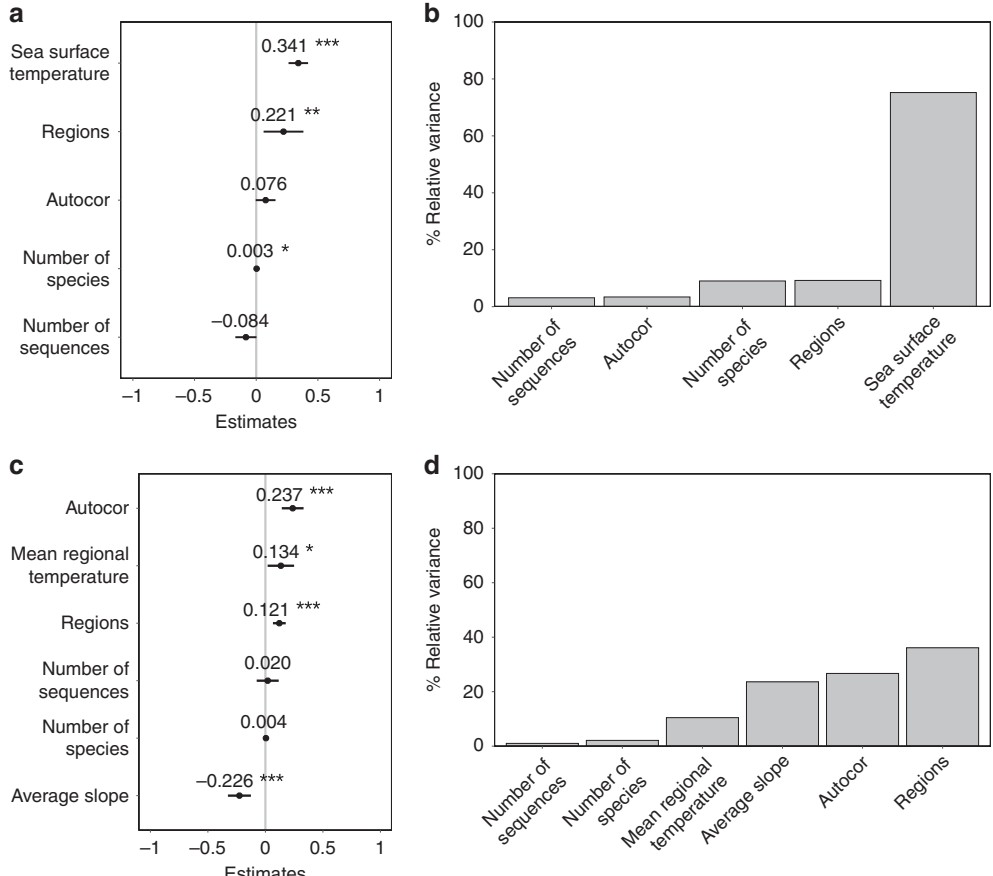

**Fig. 3 Determinants of fish genetic diversity patterns.** Outputs of the linear models (lm) testing the effect of geographic, environmental and sampling factors on the global pattern of marine (**a**, **b**) and freshwater (**c**, **d**) genetic diversity (See Supplementary Table 3 and 5 for details on models). Coefficients and confidence intervals for the factors of the models for marine (**a**) and freshwater fishes (**c**). Confidence intervals were estimated from the standard error of each coefficient at a level of 5% and were obtained with the command *confint* in the R package *lm*. Autocor is a spatial autocovariate that takes into account spatial autocorrelation in both our predicted and predictive variables. Relative variance of genetic diversity explained by the various factors was estimated and represented as partial plots with the package *hier.part* in marine (**b**) and in freshwater fishes (**e**).

underlying processes especially those linked to temperature for marine species (Fig. 3a, b, Supplementary Fig. 3). For freshwater fishes, we highlight marked contrasts in diversity across geographic regions as well as a strong influence of the average slope of river basins (Fig. 3a, b). These latter results suggest that unmeasured region-specific properties (e.g. biogeographic history), but also characteristics related to the shape of the river basins might shape the distributions of genetic diversity in these ecosystems.

For marine species, the energy, evolutionary speed and colonisation hypotheses are all candidate mechanisms to explain the positive relation between genetic diversity and sea temperature[38,39]. The higher fish genetic diversity in warmer cells could result from the positive effect of energy on population sizes[25,26]. However chlorophyll a, as a surrogate for productivity[40], was not retained in the final model and was weakly and negatively correlated to temperature ($r = -0.10$, $p = 0.018$). This result suggests that its effect on genetic diversity through population size, if any, is masked by other factors, such as temperature stability. In contrast, our results are consistent with the evolutionary speed hypothesis, which posits that warm temperatures shorten generation times and speed-up mutation rates thus, potentially increasing genetic diversity[41]. Our results are also consistent with the colonisation hypothesis and with the idea that past demographic events can shape the current global patterns of marine genetic diversity. For example, in the Indo-pacific,

climatic stability buffered the effect of Quaternary climatic fluctuations on species extinction[4] and might have limited genetic diversity erosion. Our study cannot disentangle the predictions of the speed and colonisation hypotheses to explain the global pattern of genetic diversity in marine fish. However, our findings represent one of the first study suggesting that temperature might be causally linked to global patterns of intraspecific diversity. All these patterns and potential effects are conserved in marine species when selecting only one-third of cells with more reliable estimations of genetic diversity (higher number of sequences or species), suggesting that our conclusions are robust to sampling bias.

Our study reveals contrasts in fish genetic diversity between freshwater and marine environments. Marine systems are known to host a lower alpha diversity globally than freshwater systems despite occupying much larger surface area[35,42]. Our analyses do not show a marked difference in median species diversity per cell (median value of 300 species for freshwater vs. 268 species for marine fishes, Supplementary Fig. 3a), but show a difference in fish genetic diversity between marine and freshwater cells (Supplementary Table 3), with freshwater genetic diversity being about twice higher than marine genetic diversity. This finding might appear surprising, since the greater connectivity in marine systems compared to rivers[27] and the high effective population sizes expected in marine fishes[43] might have sustained genetic diversity at least as large as that observed in freshwater. It is

noteworthy that at a given latitude, habitat difference (freshwater vs. marine) explains 67% of partial variation in genetic diversity (Supplementary Table 3). Every measure of diversity has a dependence on the spatial scale. Our results reflect the size of the cells used for this study (200 km). At this spatial scale, the difference in genetic diversity between marine and freshwater species is most likely explained by the different horizontal physical complexity of marine vs. freshwater environments since we do not observed any effect of bathymetry (marine species) or altitude (freshwater species). In particular, freshwater systems are often considered as island-like in which evolutionary dynamics are driven by structural components such as the river or lake network complexity and by the steep physical gradients over relatively short geographical distances[14,44]. The horizontal habitat complexity may promote genetic diversity in these ecosystems much than in marine systems. Moreover, this habitat complexity may also promote genetic diversity much more than species diversity, because it can favour the persistence of isolated populations within species in a very restricted area, but not the coexistence of a large number of species in competition within the same water segment.

We highlight an association between temperature and genetic diversity in the marine environment. A positive association between temperate and genetic diversity, was also observed in freshwater species, although less marked than in marine species. Freshwater fish genetic diversity was, conversely, mainly associated with the region and the average slope of river basins (Fig. 3). In more homogenous and connected marine systems, large-scale environmental gradients such as sea surface temperature can play a dominant structuring role as regard to genetic diversity. In contrast, in freshwater systems, landscape characteristics might have more pronounced effects on the spatial patterns of genetic diversity than environmental gradients[14,42,45]. The negative relationship between the average slope of river basins and genetic diversity is theoretically expected given that steeper rivers are characterised by less stable hydrological conditions[46] and, hence, lower population sizes and lower genetic diversity. Moreover, steeper rivers are more difficult to reach and have hence been less prone to rapid re-colonisation after the last glaciation, which also tends to decrease genetic diversity[14,47]. The lack of association between total basin area and genetic diversity combined with the fact that genetic diversity is negatively associated to the river slope suggests that colonisation processes might be more important than contemporary population sizes in explaining patterns of genetic diversity in freshwater systems, as previously suggested by studies at the river basin scale[14].

Intraspecific genetic diversity in freshwater systems shows a strong biogeographic signal, with the highest level found in South America, a region also supporting the highest species diversity[48]. The high freshwater fish diversity observed in South America has been attributed to the presence of large and complex river systems owing to the species-area relationship and spatial habitat complexity[35], but also to the high availability of energy, which reduces species extinction rate, and to historical contingencies[3,48–51]. Interestingly, the diversity-area hypothesis is unlikely to explain freshwater fish genetic diversity distribution since basin area was not detected as a predictor of freshwater fish genetic diversity. The congruence of the high level of intra- and interspecific diversity in that region suggests that the complex river system of South America might promote diversity at both micro and macroevolutionary scales and reinforces the idea that this area is a major hotspot for multiple biodiversity facets and taxonomic groups[52,53].

In summary, the physical habitat complexity, the evolutionary speed and the colonisation hypotheses are likely the best candidates to explain global patterns of genetic diversity in freshwater

species. However, the weakness of the relationship between genetic and species diversity also indicates that the processes underlying genetic diversity patterns might not be completely similar to those underlying species diversity patterns. These differences might be explained by disparities in temporal and spatial scales or in responses to environmental changes at which parallel ecological and evolutionary processes operate (mutation vs. speciation; genetic vs. ecological drift; gene flow vs. dispersal; selection vs. environmental filter)[54].

Although spatially extensive, our study has some limitations associated with the data mining and the analyses performed. The samples used in Fig. 1 represent only 26% of marine and 11% of freshwater species worldwide, but cover globally 100% of fish orders and 70% of families (Supplementary Fig. 5; Supplementary Table 9). When one-third of the cells with at least two species and more than four sequences for marine species and five sequences for freshwater species were filtered out, the taxonomic coverage of our full dataset only decreased to 95% for orders and 61% for families. With this stringent filter based on minimum absolute number of sampled species and consistently high global taxonomic coverage, the correlations between genetic and species diversity increase and the significance of the main parameters remains nearly significant (Supplementary Table 7). In all other cases, the global taxonomic coverage (of our full dataset) is largely maintained (98% of orders and 69% of families covered). The dataset has some major gaps across the globe, which might influence the estimation of associations between genetic diversity and environmental factors (Supplementary Fig. 6). For example, the African realm has been largely under-sampled compared to other realms (Supplementary Fig. 6). Sampling efforts should be expanded to improve the global coverage of biodiversity information at the intraspecific level[18,31]. Moreover, we only focus on a small barcode representing mostly neutral genetic variation, while genetic diversity should be best studied across the whole genome[55]. The explanatory power of statistical models is relatively low. A large part of the unexplained variation can be due to noise in the data at different levels (e.g. limited sample size within each cell, unbalanced species representation). The remaining variation might be due to a lack of factors that may contribute to better explain variations in genetic diversity (e.g. prevalence of ecological strategies). In addition, past demography history can also have a strong effect on current genetic patterns and is probably not fully integrated into the models, or absorbed by the regional structure. To partially account for missing underlying factors, we used a spatial autocovariate at a resolution of ~200 km, which is reasonable given the large scale of our analyses[56]. We used sea-surface temperature to explain patterns of genetic diversity for fishes while some inhabit deep waters. They represent ~40% of fish species (bathymersal, bathypelagic and demersal species) potentially not living close to the surface among the 3815 marine species considered in our study. Yet, sea surface temperature is strongly correlated with sea bottom temperature at the scale of continental shelf (0–200 m)[57]. In addition, in our case, if sea surface temperature was a poor predictor for those species living in deeper seawaters, we should have detected an effect of the distance to the shore in the model. However, this factor was not shown to significantly influence mean genetic diversity per cell.

In conclusion, by adding a spatial perspective, our study provides one missing piece of the current debate on the global determinants of genetic diversity that was lacking in recent works[5,55], but see[18]. The positive, albeit weak, association between genetic and species diversity can facilitate the conservation of both biodiversity components where those metrics are congruent[58]. Conversely, more conservation challenges arise where those metrics are not congruent[59,60]. As fish genetic

diversity shows a spatial signature, this information can be used to frame conservation actions at global scale to maintain multi-layered biodiversity in the Anthropocene[15] and to better guide future attempts to mitigate the impacts of global changes on vulnerable aquatic vertebrates[61]. Therefore, genetic diversity, as well as functional and phylogenetic diversity, should be taken into account when protecting biodiversity in its broadest sense, and this complexity requires a multifaceted framework in conservation[62]. Future studies using recent population genomics approaches (such as pool-sequencing[63]), which allow for the sequencing and/or genotyping of thousands of molecular markers (such as SNPs) evenly spread over the entire genome (both nuclear and mitochondrial genomes) in both neutral and adaptive regions should greatly improve our understanding of large-scale patterns of genetic diversity over space, time and taxa. Such studies would tell us how past demographic events (fluctuation in effective population size, substructure, migration) as well as genetic processes such as mutation, selection, gene flow or drift have shaped genomes in space and time[64], in order to ultimately better inform conservation strategies under ever more fluctuating and uncertain conditions.

## Methods

**Georeferenced sequence collection.** Following on from Miraldo et al.[18], we collected mitochondrial "actinopterygii" gene sequences BOLD (http://www.boldsystems.org at 09/17/2018) using customised scripts. We only kept sequences with species, coordinates (latitude, longitude) and region name information. Sequences with the region information were georeferenced using GeoNames.org (http://api.geonames.org). This tool assigns GPS coordinates to locality names. We removed sequences with IUPAC ambiguity. We kept sequences longer than 500 bp and annotated as Cytochrome Oxidase Subunit 1–5′ Region (CO1). Species with only one sequence were also removed. The number of sequences and species retrieved at each filtering step is reported in Supplementary Table 1. We collected a total of 58,565 CO1 sequences from 5,912 actinopterygii species (Supplementary Table 1).

**Mapping fish genetic diversity.** Mean fish genetic diversity per cell was estimated across all species and mapped on a grid cell covering the study area for marine (Fig. 1a) and freshwater species (Fig. 1c). We created the worldwide grid at 200 km resolution using an equal area Behrmann projection. We selected the 1299 cells containing sequence collection by making an intersection between the grid and the coordinates of the sequence using the gIntersects function from the rgeos package in R. Multiple sequence alignments of the 58,565 COI sequences of the 5912 species using MUSCLE3 were performed to estimate genetic diversity per cell[65]. The alignments were checked manually using the software ugene[66]. In addition, only pairwise alignments with overlap >50% were kept to calculate genetic diversity. Aligned sequences were separated out for the freshwater (1781) and marine species (4131). The list of marine species was extracted from fishbase[67]. We assigned each of these aligned sequences to its cell in the grid based on its coordinates. We calculated the genetic diversity per cell following Miraldo et al.[18]. Based on its geographic coordinates, each sequence is assigned to a cell on the grid. For each cell, we estimated the nucleotide diversity (Π) of each species as the average number of variable sites in each pairwise sequence comparison following Eq. (1)[68].

$$\Pi = \frac{1}{\binom{n}{2}} \sum_{i=1}^{n-1} \sum_{j=i+1}^{n} \frac{k_{ij}}{m_{ij}} \quad (1)$$

where $k_{ij}$ the number of nucleotides that are different between sequence $i$ and sequence $j$. $n$ is the number of sequences and $\binom{n}{2}$ is the number of possible pairwise comparisons, and $m_{ij}$ is the number of shared base pairs between sequence $i$ and $j$. We estimated the genetic diversity (GD) in each cell of the grid as the mean of all species nucleotide diversities averaged across species.

**Mapping fish species diversity.** Marine fish species data were obtained from the Ocean Biogeographic Information System (OBIS, http://www.iobis.org). We inventoried 16,238,200 occurrence records from 34,883 entries. We cleaned the data by identifying synonyms, misspellings and rare species (only one occurrence) and restricted them to species present in the marine environment according to FishBase[67]. We reconstructed distribution maps for each species, defined as the convex polygon surrounding the area where each species was observed. The resulting polygon was divided into four parts across the world to integrate possible discontinuity between the two hemispheres and the Atlantic and Pacific Oceans. For example, antitropical species are distributed in northern and southern

hemisphere, but show range discontinuity near the tropics[69] and a polygon division can account for this singularity. We refined each species distribution map by removing areas where maximum depths fell outside the minimum or maximum known depth range of the species[70]. As the OBIS database does not properly represent tropical fish assemblages, we merged this database with the Gaspar database at 1° resolution that encompass 6316 coral reef species[4]. We obtained a world database containing most marine fish species that we aggregated on a 1° resolution grid covering all oceans, as this resolution is useful for other projects. A freshwater dataset was obtained from the occurrence data provided by Tedesco et al.[71]. We first compiled the polygon of each available species from the occurrence table, and then aggregated this information into a presence-absence matrix. We obtained a world database containing freshwater fish species on a 1° resolution grid covering all terrestrial parts of the Earth. Considering the scale and resolution of the current study (200 km), our polygons represent the distribution of species with sufficient accuracy compared to those commonly used in macroecological studies[72,73].

For marine and freshwater species, genetic diversity per cell was aggregated by latitude bands of 10°. We then plotted the genetic diversity per band of latitude using R. The confidence interval for genetic diversity by latitude band (standard deviation of the genetic diversity per cell divided by the square root of the number of cells) was reported in the plot representing the variability of genetic diversity at latitudinal bands amongst cells (Fig. 1b, d). We also reported a grey gradient to indicate the number of cells used in each latitudinal band (Fig. 1b, d).

**Statistical analyses.** We tested the Pearson correlation between fish genetic and species diversity using a linear model. However, as these spatial variables were observed over the same locations, we tested the significance of the association using the modified $t$-test of spatial association (function modified.ttest; R package spatialpack)[74]. To test for the difference between marine and freshwater diversity (both at the intra- and interspecific levels respectively) while accounting for latitudinal variation, we applied a linear model between genetic (or species) diversity and two factors: latitude and a binary variable indicating whether the sequences are from marine (1) or freshwater (0) fishes. Diversity metrics were log-transformed and standardised to produce variables following a normal distribution before the linear regressions. Genetic Diversity (GD) is, in theory, a proportion. However, in practice it takes only small values (Supplementary Fig. 1b, d). Therefore, previous transformations of GD produced a variable following a normal distribution.

To explore the determinants of fish genetic diversity (GD), we used linear models to test the influence of geographic ($Z$), environmental ($Y$) and sampling ($S$) factors (Supplementary Table 4). All analyses were conducted independently for marine and freshwater species. The GD variable was again log-transformed and standardised for all statistical models using the scale R function to produce a normal variable. The geographical factors included regions, bathymetry and distance to shore that increases with ocean depth for marine species only, and elevation, basin area, slopes (average and range) and flow accumulation for freshwater species only (Supplementary Table 4). Basin area has the same value for all the cells from a given basin, whereas flow accumulation provides a local cell estimation of the watershed size, with upstream cells having lower values than downstream cells. Theoretically, basin area should correlate positively with genetic diversity since higher regional (basin-scale) effective population sizes should be supported in larger river basins. An effect of the basin area reflects regional-scale processes and can be interpreted for instance in terms of past history (e.g. founder effects due to past colonisation) or connectivity at the scale of the basin. We are also expecting a positive correlation between flow accumulation and genetic diversity through processes acting at the local scale; for instance, higher flow accumulation suggests higher local effective populations sizes and hence higher genetic diversity, irrespective of the basin area. For slope-related variables, a negative relationship between the average slope of river basins and genetic diversity is theoretically expected given that steeper rivers are characterised by smaller and less stable hydrological conditions[46], and that steeper rivers might have been less prone to post-glacial colonisation[47]. This habitat instability is higher for large values of range slope. The environmental variables included temperature (sea surface temperature for marine species and air temperature for freshwater species), and, for marine species oxygen concentration and chlorophyll-a (Supplementary Table 4). The variable chlorophyll-a is theoretically associated with higher productivity and hence higher population size. The environmental variables were standardised before analysis. Regions were defined as Atlantic vs Indo-pacific for marine species and Africa, Antarctica, Europe, North America, Oceania and South America for freshwater species. We also included the effect of sampling in all models (number of sequences and number of species in each cell of the grid). Before any calculations were made, we checked for factor collinearity in the model using a variance inflation factor (VIF) procedure. Highly collinear variables with a VIF >5 were removed from the model.

Both the environmental factors and the diversity metrics inevitably show some spatial autocorrelation. We investigated the spatial autocorrelation in genetic diversity with a Moran spatial autocorelogramme using the R function pgi.cor (package pgirmess). We built an autocovariate variable for the fish genetic diversity metrics (function autocov_dist in spdep R package) to account for spatial dependency and estimate how any cell reflects the values of the neighbouring cells[75]. This autocovariate reflects variations at the geographic resolution of and

above 200 km and accounts for most of the spatial dependence in fish genetic diversity not explained by the selected factors. However, since this autocovariate was unconditional to the environmental variation (i.e. the response variable could show a spatial autocorrelation because the environment is itself autocorrelated), we regressed each autocovariate variable against the whole set of environmental factors using a linear model. We extracted the model residuals and then used them as a spatial variable independent of the environmental factors considered in the study to predict fish genetic diversity. This new spatial variable (i.e. model residuals) was called autocor in our analyses.

We then applied a stepAIC regression procedure to each linear model and considered the model with the smallest AIC as the best, i.e. most parsimonious, model. The sampling factors ($S$) were forced in all models, even those which were not significant since they account for data structure. Relative variance of genetic diversity explained by the various factors was estimated and represented as partial plots with the package *hier.part*. We visually checked the independence and normality of the residuals of all models. In addition, we performed a spatial analysis of the residuals (Moran $I$ test; 1000 permutations considering the nearest neighbour value (R function moran.mc, $k = 1$)) to test whether they are not spatially autocorrelated ($H_0$) indicating that the model is not missing a major variable with a spatial structure and that the coefficients of explanatory variables are not biased. It represents an a posteriori analysis evaluating whether the model outputs are reliable.

We finally investigated the influence of more stringent thresholds for absolute species or sequences number and taxonomic coverage to estimate genetic diversity per cell and thus test the robustness of our findings to some arbitrary choices (number of species and sequences per cell and taxonomic coverage per cell). We choose thresholds that allowed to select around one-third and two-third of the "best" grid cells when possible (Supplementary Tables 7 and 8). The taxonomic coverage used for this analysis is mapped in the Supplementary Fig. 7.

**Reporting summary**. Further information on research design is available in the Nature Research Reporting Summary linked to this article.

## Data availability

The datasets generated during and/or analysed during the current study are available in the worldmap_fish_genetic_diversity repository, https://gitlab.mbb.univ-montp2.fr/reservebenefit/worldmap_fish_genetic_diversity. The source data underlying Fig. 1b, d, 2a, c, 3a, c and Supplementary Figs. 1 to 7 are provided as a Source Data file.

## Code availability

All the scripts to reproduce the results are available in the worldmap_fish_genetic_diversity repository: https://gitlab.mbb.univ-montp2.fr/reservebenefit/worldmap_fish_genetic_diversity.

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

## Acknowledgements

We thank Michel Kulbicki for his helpful input. This work benefited from the Montpellier Bioinformatics Biodiversity platform supported by the LabEx CeMEB, an ANR "Investissements d'avenir" programme (ANR-10-LABX-04-01). P.E. is funded by 2015 −2016 BiodivERsA COFUND call for research proposals, with the national funders ANR (France). We would also like to thank the scientists who collected the data and generated the sequences used in the meta-analysis. L.P. was supported by the SNF project REEFISH no. 310030E-164294.

## Author contributions

S.M. and L.P. designed the study. L.V. and P.E.G. extracted the data and designed the spatial grid. P.E.G., S.M. and D.M. co-analysed the data. P.E.G. produced the workflow of all scripts. P.E.G. and C.A. produced the figures. S.M. produced a first draft of the paper. S.M., P.E.G., D.M., S.B., L.V., C.A. and L.P. contributed to the writing and valuable scientific interpretations.

## Competing interests

The authors declare no competing interests.
