## [Peer Review File · Nature Communications]

Reviewers' Comments:

Reviewer #1:

Remarks to the Author:

This study assembled and geo-referenced the largest dataset of DNA sequences for marine and freshwater fish species to estimate global scale patterns of intra-specific genetic diversity and unveil the drivers of those patterns. Results reveal a latitudinal pattern of intra-specific genetic diversity, with tropical regions overall hosting the higher levels of genetic diversity. This is in my knowledge the first time a global pattern of genetic diversity in fishes based on an extensive dataset and therefore it contributes to reveal the last unknown global pattern of fish biodiversity (richness and ecosystem diversity are the other two and have been described before). Results show significant correlation between genetic diversity and mainly temperature for marine fishes, and a significant role of region for freshwater genetic diversity.

I highly appreciate the challenge that authors have faced to assemble and curate the database, and the study illustrates well the potential of analysing large scale patterns of genetic diversity. I am however concerned with some aspects of the analysis and of its overall interpretation.

First, the explanatory ability of predictors, even significant, is low (with explained variation lower than 20%). What may explain the other 80%? Are values below 20% enough to claim a strong control, for example of temperature, over the spatial patterns of genetic diversity? I would suggest authors to down-tune their claims on the current ability of their predictors to explain genetic diversity and maybe briefly discuss why those values of explained variance are still low?

Second, I would suggest authors to explore the impact of the amount of genetic data and taxonomic coverage on their result. I believe that if authors would select the top tier of their grid-cells (i.e., the best 200 grid-cells), the degree of support to their hypothesis may well improve as a result of better data. Estimates of GD have been shown to be sensitive (and vary largely) when the amount of data and taxonomic coverage within each grid cell is low (Miraldo et al. 2016).

Third, the study would benefit from an extended description of protocols and decisions behind the alignment of sequences. I miss information for example on the degree of minimum overlap among sequences, for example, to rule out alignment sequences that do not significantly overlap.

Fourth, are sea-surface temperatures relevant for fishes inhabiting deeper waters? What proportion of their marine fish species may be directly affected by sea-surface temperatures? May be species inhabiting deeper water that may not be affected directly by sea-surface temperatures? Please discuss.

Fifth, readers would benefit of a description of the taxonomic coverage of the study. Are some groups of fishes (family or genera) over-represented or under-represented? What that would imply for your results?

There is also a list of minor comments which may be useful for the authors:

- a) You mention both GenBank and BOLD as sources of information...please clarify
- b) Grid cell size should be reported in the main text. Helps the reader to better apprehend the spatial scale of the analysis without the need to find it in methods or supplementary materials.
- c) Lines 136-139 report spatial patterns again in the same paragraph that mechanisms. Maybe move

to the parts of the text describing spatial patterns?

d) Line 252. Authors wrote "... 10 individuals per species (IC = [9.43, 252 10.34])". Do you mean individuals or just sequences? How do we know that each sequence represents a different individual? Maybe more than one sample have been sequencing for the same individual?

e) Why the grid-cell size for species richness is at 1 degree and genetic diversity at 200 kms? Would not be more adequate to aggregate richness at the same resolution that GD?

Reviewer #2:

Remarks to the Author:

This is a very interesting subjects, and the manuscript is well-written, however, I don't think the data is quite there yet.

Introduction: There are quite a few hyperboles here. Scientists are not "neglecting" the study of genetic diversity at a global scale. The entire field of phylogeography is quite new, and there simply has not been enough data out there for many studies like these yet.

Lines 75-88: The introduction to fish diversity is misleading. Fish diversity is not distributed evenly among the oceans, instead, it is highly concentrated on the coastal areas. So coastal ecosystems (which occupy an area similar to freshwater ecosystems) are where marine fish diversity is similar to freshwater, and the open ocean (which comprises the vast majority of the ocean)

Line 84: Again, this is not surprising since there hasn't been enough datasets published for a robust analysis. You just happen to be the first.

Results: Even here I am not sure you have enough data to support your conclusions. You analyzed 41462 and 17103 sequences from 4131 marine and 1781 freshwater species. That's (amazingly for both systems) on average 10 sequences per species. Is this enough to determine genetic diversity of an entire species? Were those 10 sequences from the same location or different locations? This could completely change your estimate of genetic diversity. In the methods you say that species with only 1 sequence were removed. I think that number should be raised at least to 5 and that might still be too low (I think your average number of 10 might be too low to reliably determine the genetic diversity of an entire species). My suspicion is that the vast majority of species will have 2-3 sequences and others (the ones that were subject of more detailed phylogeographic studies) will have 100s. But even if that's not the case, your average genetic diversity per species should be accompanied by a standard error and those not significantly different from zero (probably the ones with just 2-3 sequences per species) should be eliminated from the analysis. When you do this, you will realize why such global comparisons were neglected in the past. And actually, your analysis clearly shows that as the only regional subdivisions you were able to make in marine fishes for example was Atlantic versus Indo-Pacific (line 307).

In your regional categorization something else jumped out as odd. You had an "Antarctica" regional category for freshwater fishes. Which freshwater fish is in Antarctica? The dots in the map of freshwater fishes in Figure 2 shows locations in the ocean in Antarctica.

In the global analysis, the peaks of high genetic diversity clearly correspond to the areas where more Barcoding studies have been done. This is demonstrated clearly in South America. Here the highest species diversity is around the equator (Amazon basin), but there are many more species sampled

across a wider geographical area in southeastern Brazil (20S), hence the higher genetic diversity further south.

The conservation link is weak. Here the authors generalize their data as "genetic diversity", however, we have to remember that what we have here is mitochondrial DNA COI diversity. And those are two separate things. A species can have very high mtDNA diversity and be on it's way to extinction, or it can have very low mtDNA diversity and be doing just fine. In addition, the time scale in which mtDNA evolution occurs often reflects population events that happened thousands to tens of thousands of years ago. The relatively low haplotype diversity (star phylogenies) in mtDNA of coastal marine fishes for example is widely accepted as being caused by the low sea level 10k years ago. So what we do to these populations now (which would affect conservation) will only show up in mtDNA hundreds to thousands of years from now.

Final suggestion: maybe thank the thousands of scientists who collected the data and generated the sequences you are using in this meta-analysis.

Reviewer #3:

Remarks to the Author:

The manuscript reports about an analysis of global patterns of mitochondrial genetic diversity (GD) in ray-finned fish, based on georeferenced cytB sequences in the BOLD database.

The manuscript is generally well written and concise (with a few exceptions). Statistical analyses are generally appropriate, although I have marked some relatively important criticism (possibly the most important: simple linear models were used, and with no explanation, for a 0 to 1 proportion response, when a beta regression would probably more appropriate).

The study's approach is not novel, as it closely follows that by Miraldo et al. 2016 (appropriately cited in the manuscript, as ref. 19), applying it to a new taxon.

Compared to Miraldo's paper, this study takes a slightly different, and interesting, angle, in that it attempts to link GD to functional ecological variables (temperature, primary productivity, oxygen) and geographical "regions", corresponding to the main oceanic basins (Atlantic vs IndoPacific for marine systems and continents for freshwater). Also, the authors made a positive effort in accounting for spatial autocorrelation in all statistical analyses.

The main claims of the study are that:

- 1) GD is geographically heterogeneous, being typically higher in tropical regions for both marine and freshwater environments
- 2) GD is positively correlated to temperature, especially in marine habitats
- 3) regional effects are marked in freshwater habitats, with South America hosting areas with highest GD
- 4) GD is higher in freshwater than marine fish
- 5) GD is positively correlated to species diversity for both marine and freshwater fish.

These results are of interest, as no similar global analysis has been proposed so far for fish. However, my impression is that they do not provide exceptionally novel insight about major biogeographical hypotheses.

For example, three well-known hypotheses explaining higher diversity in the tropical regions

("evolutionary speed", "colonization", "energy-richness") are presented in the Introduction, and, in the Discussion, it is stated that the data are consistent with all of them. Indeed, as stated in the aims, the study was not specifically designed to do distinguish among these hypotheses, so that the whole argument appears not relevant. Consequently, however, the observed correlation of GD and temperature remains of relatively little explanatory power (similarly to the - weak - correlation of GD and species richness). I am particularly perplexed by the quick dismissal of a promising analysis on ocean productivity on the ground that chlorophyll is highly correlated with ocean temperature. I am surprised that there is a high and *positive* correlation between marine productivity and temperature (the argument that "the positive relation detected between genetic diversity and temperature can also be explained by productivity", of course, only makes sense if this correlation is positive!). I am not an expert, but it seems to me that the global pattern is a *negative* correlation. This should be better discussed. Also, the oxygen predictor is mentioned in the methods but disappears without any comments.

Result 4 seems interesting to me. However, It would deserve some improved analysis/discussion. In fact, if, as speculated by the authors, the higher GD among freshwater fish depends on the higher structural complexity of freshwater environment, we might expect a marked dependence on the geographic scale, with freshwater fish expected to harbour lower diversity within each water body but higher diversity within each cell of an appropriately scaled grid. It seems to me essentially similar to measuring alpha and beta diversity in ecology. I wonder if a more detailed analysis would not be possible using the same databases and some more geographic data (e.g., watersheds and river basins or exploring different grid sizes?). Similarly, the interpretation given for the differences among continents (that, e.g. South America hosts large river basins with high structural complexity), could have been straightforwardly tested by assigning cells to river basins and accounting for their size.

All in all, I suggest that the study may be improved by correcting a few statistical inaccuracies and, importantly, by attempting some more careful analysis/discussion of relevant geographic variables (e.g. size of river basins, river network complexity, actual distance among sampled conspecific individuals within cells, marine productivity). Provided that, I think it has the potential to be published in Nature Communications.

I have marked more detailed comments (including some relatively important remarks on statistical methods) in a commented manuscript file.

Paolo Gratton

Detailed answers to Reviewers' comments:

Reviewer #1 (Remarks to the Author):

First, the explanatory ability of predictors, even significant, is low (with explained variation lower than 20%). What may explain the other 80%? Are values below 20% enough to claim a strong control, for example of temperature, over the spatial patterns of genetic diversity? I would suggest authors to down tune their claims on the current ability of their predictors to explain genetic diversity and maybe briefly discuss why those values of explained variance are still low?

We agree that the explanatory ability of predictors is rather low. As recommended by the reviewer, we have added discussion of the low explanatory power of our model in the revised version and have also toned down our claim in the main text. The new discussion reads 1447-524 :

“Fourthly, the explanatory power of our factors is relatively low (16% for marine species). This might be due to a lack of factors that may contribute to better explaining variations in genetic diversity. However, at this scale, information on habitat, human impact or connectivity is either not available or of poor quality. In addition, past demography history can also have a strong effect on current genetic patterns and is probably not fully integrated into the spatial variation. In order to partially account for these missing variables, we used a spatial autocovariate at a resolution of approximately 200km, which is reasonable given the large scale of our analyses. This accounts for the spatial dependence among environmental factors. The resolution considered here is not appropriate for investigating within-cell spatial autocorrelation.”

Second, I would suggest author to explore the impact of the amount of genetic data and taxonomic coverage on their result. I believe that if authors would select the top tier of their grid-cells (i.e., the best 200 grid-cells), the degree of support to their hypothesis may well improve as a result of better data. Estimates of GD has been shown to be sensitive (and vary largely) when the amount of data and taxonomic coverage within each grid cell is low (Miraldo et al. 2016).

We thank the reviewer for this comment. Firstly, we now quantify the taxonomic coverage of our dataset: we have provided a new figure in the supplementary information (S6) and a new table (Supplementary Table 7) describing the number of species and sequences per taxa and per family, and we have described the taxonomic coverage in the discussion. In brief, the dataset used in our model covers all orders and 70% of the families when using all grid cells in the model. When we decreased the number of cells (see below for explanation), we still covered 95% of orders and 61% of families. We have added this information to the discussion 1430-438.

Secondly, we investigated the influence of the amount of genetic data on our results. We increased the number of sequences or the species diversity in each cell of the grid and only kept grid cells with top values (about 1/3 and 2/3). We then estimated the regression coefficients of the model reported in Table S5 and quantified the variation among the regression coefficients in our model and the coefficient in the new models.

For marine species, when the minimal number of sequences per species (4) or the minimal number of species per cell (8) is increased, the pattern is preserved and temperature and region are still significant (Supplementary Table 6).

For freshwater species, we observed the same changes except when we filtered for 5 sequences. In this case, temperature was no longer significant. Overall, this shows that our findings are relatively robust to changes in the databases and to data quality, which reinforces our conclusions.

We have added this information to the method 1 700-704 and the results section 1295-327 of this revised version of the MS. The suggestion of the reviewer was an excellent one since we obtained greater explanatory power in our models (higher adjusted R^2) when less cells were kept with a more reliable estimation of genetic diversity in each cell. This supports the detected patterns and our assumptions. We have added this information to the discussion 1403-405 (“All these patterns and effects are conserved when selecting only one-third of sites with more reliable estimations of genetic diversity, suggesting that our conclusions are robust to the sampling design.”)

Third, the study would benefit from an extended description of protocols and decisions behind the alignment of sequences. I miss information for example on the degree of minimum overlap among sequences, for example, to rule out alignment sequences do not significantly overlapping.

We have added details about the protocols (see l. 550-552). The alignment of sequences was checked visually as in Miraldo et al. (2016). The alignments were checked manually using ugene. Additionally, only pairwise alignments with overlap greater than 50% are kept to calculate genetic diversity.

Four, are sea-surface temperature relevant for fishes inhabiting deeper waters? What proportion of their marine fish species may be directly affected by sea-surface temperatures? May be species inhabiting deeper water that may not be affected directly by sea-surface temperatures? Please discuss.

Good point. Sea surface temperature can appear to be a poor factor for bathydemersal, bathypelagic and demersal species that inhabit deeper waters. Using this classification, we obtained a proportion of 40% of species potentially not living on the surface among the 3,815 marine species considered in our study.

Yet, Sea Surface Temperature (SST) is commonly used to model marine species distribution in coastal ecosystems (Albouy *et al.* 2013; Cheung *et al.* 2009; Lasram *et al.* 2010; Cheung *et al.* 2013; Saupe *et al.* 2014). The main reason for this is that SST is strongly correlated with sea bottom temperature (SBT) at the scale of the continental shelf (0-200m). As an example, Cheung et al (2013) show the following relationship (Fig. 1) between modelled sea surface temperature and sea bottom temperature in 2000 from NOAA’s GFDL Earth System Model 2 for shelf seas (0.200 m depth) in the North Atlantic.

Figure 1: Relationship between modelled sea surface temperature and sea bottom temperature in the year 2000

The result is similar for the Mediterranean Sea (Fig. 2). We drew the temperature anomaly relative to the 1961-1990 average to better visualise the difference between the sea surface temperature and the 150 m layer for the NEMOMED8 model.

Figure 2: Temperature anomaly relative to the 1961-1990 average between the sea surface temperature (in green) and the 0-150 m layer (in blue) for the NEMOMED8 model.

We can conclude that the water warming and the variation in water temperature is similar between the two layers. Finally, in our case, if SST was a poor predictor for those species living in deeper sea, we should have detected an effect of the distance to offshore in the model. However, this variable was not shown to significantly influence the genetic diversity per cell. Additionally, sea surface can be used as a proxy of the deeper sea temperature here as we are interested in the spatial variation in the signal of temperature. We decided to not add this multi-argument discussion in the paper.

Fifth, readers would benefit of a description of the taxonomic coverage of the study. Are some group of fishes (family or genera) over-represented or under-represented? What that would imply for your results?

We have now added a supplementary Table 7 and a supplementary Figure 6 to describe the taxonomic coverage. We also discuss this point l 460-432 “the data used for the global maps (Fig. 1) and models (Fig.2) covered 100% of fish orders and 70% of families (Supplementary Figure 5; Supplementary Table 7)”

There is also a list of minor comments which may be useful for the authors:

a) You mention both GenBank and BOLD as sources of information...please clarify
This is a mistake. We used BOLD. We have removed references to GenBank throughout the MS.

b) Grid cell size should be reported in the main text. Helps the reader to better apprehend the spatial scale of the analysis without the need to find it in methods or supplementary materials.

We have added this information l 161-163: “We then estimated nucleotide diversity per site for each species in each grid cell at a resolution of 200km worldwide”.

c) Lines 136-139 report spatial patterns again in the same paragraph that mechanisms. Maybe move to the parts of the text describing spatial patterns?

Sorry for the confusion. The significant effect of the region is an output of the model, so we cannot move it. We just forgot to declare regions as a factor (Supplementary Table 3).

d) Line 252. Authors wrote "... 10 individuals per species (IC = [9.43, 252 10.34])". Do you mean individuals or just sequences? How do we know that each sequence represents a different individual? Maybe more than one sample have been sequencing for the same individual?

Individual is correct since BOLD only allows one sequence to be deposited per individual. However, given that errors in BOLD are possible we have now replaced individual per sequence throughout the paper. Thank you for this remark.

e) Why the grid-cell size for species richness is at 1 degree and genetic diversity at 200 kms? Would not be more adequate to aggregate richness at the same resolution that GD?

The resolution of 1 degree for species richness is the raw data resolution that we used for our analyses. Then species richness was averaged in each cell of 200km from the 1° resolution grid. We have added this information to the main text l631

Reviewer #2 (Remarks to the Author): This is a very interesting subjects, and the manuscript is well-written, however, I don't think the data is quite there yet.

Introduction: There are quite a few hyperboles here. Scientists are not "neglecting" the study of genetic diversity at a global scale. The entire field of phylogeography is quite new, and there simply has not been enough data out there for many studies like these yet.

We fully agree with the reviewer. This is partly due to the difficulty and the cost to sample and genotype enough individuals within species at global scale. We have re-written (l 43-47) the sentences as follow: "However, most global studies are investigating biodiversity at species level 1, 2, 3, 4, with few studies examining the diversity of genes within organisms also referred to as genetic diversity. The cost of sampling and genotyping a sufficient number of individuals within species at global scale has limited our understanding of the determinants of intraspecific genetic diversity.

Lines 75-88: The introduction to fish diversity is misleading. Fish diversity is not distributed evenly among the oceans, instead, it is highly concentrated on the coastal areas. So coastal ecosystems (which occupy an area similar to freshwater ecosystems) are where marine fish diversity is similar to freshwater, and the open ocean (which comprises the vast majority of the ocean)

We have modified the introduction as follow (l153-157):

"Marine environments cover 70% of Earth, but marine fish species diversity is mainly concentrated in coastal areas (at depth less than 200m) which represent less than 1% of the world's oceans. This overrepresentation of species in coastal areas and rivers suggests that habitat is key to fish diversity"

Line 84: Again, this is not surprising since there hasn't been enough datasets published for a robust analysis. You just happen to be the first.

We agree. We have removed "surprisingly" and have added a sentence to the introduction to describe the difficulty of sampling for genetic spatial analyses l46-47. "The cost of sampling and genotyping a sufficient number of individuals within species at global scale has limited our understanding of the determinants of intraspecific genetic diversity"

Results: Even here I am not sure you have enough data to support your conclusions. You analyzed 41462 and 17103 sequences from 4131 marine and 1781 freshwater species. That's (amazingly for both systems) on average 10 sequences per species. Is this enough to determine genetic diversity of an entire species? Were those 10 sequences from the same location or different locations? This could completely change your estimate of genetic diversity. In the methods you say that species with only 1 sequence were removed. I think that number should be raised at least to 5 and that might still be too low (I think your average number of 10 might be too low to reliably determine the genetic diversity of an entire species).

Our explanations were not clear enough in the previous version of the manuscript. We have now clarified that the unit of our analysis is the cell. We firstly estimated the nucleotide diversity for each fish species in each site/cell of the grid. We then averaged the resulting nucleotide diversity in each site/cell across species providing a mean nucleotide species diversity per site/cell. See 1 168-170 for clarification. Therefore, we do not have one estimation of nucleotide diversity per species but one per species and per cell. The 34,782 and 15,806 sequences from 3,815 marine and 1,611 freshwater fish species (and an average of 10 sequences per species), respectively, represent the total number of sequences and species used in the analyses for all locations. However, we cannot say whether we have enough data to determine the genetic diversity of an entire species or not, as this was not the aim of our analysis. We have removed the mean number of sequences for each species since it is not useful in our study.

That said, we did test the effect of increasing the minimum number of sequences for each species in each grid cell. We cannot use values higher than 4 for marine species and 5 for freshwater species since we would lose too many cells to run the model at global scale. We have discussed this sensitivity analysis in a previous comment and report the results in the supplementary Table 6.

When selecting at least 4 sequences for each species, the patterns are maintained. We note that Miraldo et al. (2016) observed the same results: patterns are conserved. Therefore, our model describing the pattern of genetic diversity in relation to different factors is robust to the number of sequences per species and the sub-selection of species. We decided to keep the results of the model with two sequences per species and two species per cell since it provides the largest number of cells which is essential when testing a global spatial pattern. The results obtained for a sub-selection of species with more sequences are provided (Supplementary Table 6).

My suspicion is that the vast majority of species will have 2-3 sequences and others (the ones that were subject of more detailed phylogeographic studies) will have 100s. But even if that's not the case, your average genetic diversity per species should be accompanied by a standard error and those not significantly different from zero (probably the ones with just 2-3 sequences per species) should be eliminated from the analysis. When you do this, you will realize why such global comparisons were neglected in the past. And actually, your analysis clearly shows that as the only regional subdivisions you were able to make in marine fishes for example was Atlantic versus Indo-Pacific (line 307).

About 30% of species have 2 or 3 sequences, and the species *Oncorhynchus tshawytscha* has 358 sequences but this is an outlier. The distribution of the number of sequences per family is given in the supplementary Table 7.

Once again, our focus is on the estimation of GD per grid cell. At the grid cell level, we can be less conservative as explained in a previous comment. We have provided a confidence interval of the estimation of average genetic diversity on the latitudinal bands in order to better characterize those estimations at the spatial level: see new Figure 2b, d. We have also provided a sensitivity analysis on the effect of removing species with only two sequences. We showed that the spatial pattern was conserved, i.e. the same important factors were selected (Supplementary Table 6) and 1303-333. We kept in the main manuscript a minimum of two sequences per species and added a discussion of the sensitivity analysis 1438-444

“When one-third of the sites with at least two species and more than 4 sequences for marine species and 5 sequences for freshwater species, were filtered out the taxonomy coverage only decreased to 95% of orders and 61% of families covered. With this stringent filter and consistently high taxonomic coverage, the model results are very similar, except for the freshwater systems in which temperature becomes non-significant. In all other case, the taxonomic coverage was largely maintained (98% of orders and 69% of families covered)”.

We fully agree that the difficulty in sampling and obtaining data on a large scale is the main limiting factor for carrying out global comparisons and we have highlighted this point in the introduction. Our conclusion calls on the scientific community to produce more global analyses for such genetic comparisons. Some of the co-authors of this paper are involved in global expeditions to collect such data and are fully aware of the costs and difficulties of large-scale sampling.

In your regional categorization something else jumped out as odd. You had an "Antarctica" regional category for freshwater fishes. Which freshwater fish is in Antarctica? The dots in the map of freshwater fishes in Figure 2 shows locations in the ocean in Antarctica.

We would like to thank the reviewer. The species in Antarctica were not freshwater species. The error was due to an annotation error in Fishbase. We detected 3 other species incorrectly assigned to the freshwater database that were removed. The number of cells used in the model has been updated after cleaning by identifying the synonyms and misspellings (Supplementary Table 2).

In the global analysis, the peaks of high genetic diversity clearly correspond to the areas where more Barcoding studies have been done. This is demonstrated clearly in South America. Here the highest species diversity is around the equator (Amazon basin), but there are many more species sampled across a wider geographical area in southeastern Brazil (20S), hence the higher genetic diversity further south.

Our model tested the effect of sampling (number of individuals and number of species). If the peaks of high genetic diversity were only in areas with more barcodes, we should detect systematically an effect of the number of sequences in the model. This was not the case since the sampling variables were not, or only slightly, significant compared to temperature or region. We agree that the map is somewhat confusing, this is why we have added models and kept sampling variables as the main factors in our analyses, to test their effects vs. the other geographic and climatic factors.

The conservation link is weak. Here the authors generalize their data as "genetic diversity", however, we have to remember that what we have here is mitochondrial DNA COI diversity. And those are two separate things. A species can have very high mtDNA diversity and be on it's way to extinction, or it can have very low mtDNA diversity and be doing just fine. In addition, the time scale in which mtDNA evolution occurs often reflects population events that happened thousands to tens of thousands of years ago. The relatively low haplotype diversity (star phylogenies) in mtDNA of coastal marine fishes for example is widely accepted as being caused by the low sea level 10k years ago. So what we do to these populations now (which would affect conservation) will only show up in mtDNA hundreds to thousands of years from now.

We fully agree that a more thorough analysis is needed with complementary data on other marker types. Whether mtDNA diversity only reflects events that occurred in the distant past and whether it reflects neutral processes or selection are questions which have been widely debated (Bohonak & Vandergast 2011; Galtier *et al.* 2009; Wang 2010). This marker has long been associated with phylogeographic studies and events that occurred in the distant past. Nonetheless, the mutation rate for this marker is high (higher than most nuclear markers) and more importantly it has a single copy, meaning that its effective population size is two times lower than any other nuclear markers. This particularity makes it relevant for studying demographic events which have occurred recently, and it can actually (sometimes) be more informative for recent events than microsatellites, for instance (and SNPs that are characterized by low mutation rate, see Bohonak and Vandergast 2011). Moreover, it is associated with neutrality, although it has also been shown that its diversity can be associated with selection and traits related to temperature such as metabolic rate (April *et al.* 2013). Finally, even if we assume that mtDNA diversity only reflects late events, it is not illogical to consider these types of markers when attempting to unravel global patterns of diversity, since the laws and rules that sustain very large spatial patterns of diversity necessarily encompass processes that occurred late in time. To sum up, identifying conservation solutions with a single type of marker is never ideal, and a combination of markers is actually the best way to proceed. We now explicitly advocate that additional data on other marker types would reinforce the type of study we have performed. Nonetheless, we are confident that our conclusions already provide useful information regarding the potential drivers of genetic diversity at a large spatial scale, and show that it is important from a conservation perspective to highlight that genetic diversity (even if from ancient processes) is poorly related to species diversity at the global scale, and that our study is a necessary step for future studies focusing on a wider set of markers (when available) which will of course refine our findings.

Final suggestion: maybe thank the thousands of scientists who collected the data and generated the sequences you are using in this meta-analysis.

Of course, we now thank the scientific community for collecting and sharing their data.

Reviewer #3 (Remarks to the Author):

The manuscript reports about an analysis of global patterns of mitochondrial genetic diversity (GD) in ray-finned fish, based on georeferenced cytB sequences in the BOLD database.

The manuscript is generally well written and concise (with a few exceptions). Statistical analyses are generally appropriate, although I have marked some relatively important criticism (possibly the most important: simple linear models were used, and with no explanation, for a 0 to 1 proportion response, when a beta regression would probably

more

appropriate).

The response variable is the genetic diversity per cell (GD), which is the average nucleotide diversity across species in each cell. The reviewer is right that the expected values of genetic diversity are proportions, and that a beta regression would be more appropriate. However, because most GD values were small (<0.28 , histograms a and c below), and because we wanted to include random terms and autocorrelation (which is not feasible using beta-regressions as far as we know), we preferred to transform the raw data to obtain a normal distribution and use a Gaussian GLM. Specifically, we first applied a log transformation and then normalized the data. The resulting variable follows approximately a normal distribution (see the histograms below after transformation of the GD values). Our choice is validated by the normality of the residuals of our model.

We have added further explanation in the revised paper 1650-652: “GD is, in theory, a proportion. However, in practice it takes only small values. Therefore, we transformed GD to produce a variable following a normal distribution. The GD variable was log-transformed and standardized for all statistical models using the scale R function”.

The study's approach is not novel, as it closely follows that by Miraldo et al. 2016 (appropriately cited in the manuscript, as ref. 19), applying it to a new taxon.

Compared to Miraldo's paper, this study takes a slightly different, and interesting, angle, in that it attempts to link GD to functional ecological variables (temperature, primary productivity, oxygen) and geographical "regions", corresponding to the main oceanic basins (Atlantic vs IndoPacific for marine systems and continents for freshwater). Also, the authors made a positive effort in accounting for spatial autocorrelation in all statistical analyses.

The main claims of the study are that:

- 1) GD is geographically heterogeneous, being typically higher in tropical regions for both marine and freshwater environments
- 2) GD is positively correlated to temperature, especially in marine habitats

- 3) regional effects are marked in freshwater habitats, with South America hosting areas with highest GD
- 4) GD is higher in freshwater than marine fish
- 5) GD is positively correlated to species diversity for both marine and freshwater fish.

These results are of interest, as no similar global analysis has been proposed so far for fish. However, my impression is that they do not provide exceptionally novel insight about major biogeographical hypotheses.

For example, three well-known hypotheses explaining higher diversity in the tropical regions ("evolutionary speed", "colonization", "energy-richness") are presented in the Introduction, and, in the Discussion, it is stated that the data are consistent with all of them. Indeed, as stated in the aims, the study was not specifically designed to do distinguish among these hypotheses, so that the whole argument appears not relevant. Consequently, however, the observed correlation of GD and temperature remains of relatively little explanatory power (similarly to the - weak - correlation of GD and species richness). I am particularly perplexed by the quick dismissal of a promising analysis on ocean productivity on the ground that chlorophyll is highly correlated with ocean temperature. I am surprised that there is a high and *positive* correlation between marine productivity and temperature (the argument that "the positive relation detected between genetic diversity and temperature can also be explained by productivity", of course, only makes sense if this correlation is positive!). I am not an expert, but it seems to me that the global pattern is a *negative* correlation. This should be better discussed. Also, the oxygen predictor is mentioned in the methods but disappears without any comments.

The previous manuscript did not include sufficient explanation on this topic which we have now added (O₂, etc.).

For marine species, O₂ is highly correlated with SST. The correlation coefficient is -0.98 ($P < 2 \times 10^{-16}$) (see the plot below on the left). Yet we detected only a slight negative correlation between chlorophyll and temperature (cor = -0.10, $p = 0.018$) (plot below on the right).

The vif procedure (vif=25) confirms that O₂ and temperature were highly correlated and we removed O₂. For bathymetry and chlorophyll-a, our threshold for the vif was too conservative in the previous version of the manuscript and we have changed it to 5 (Sheather, Simon 2009).

A modern approach to regression with R. New York, NY: Springer). We then kept bathymetry and chlorophyll-a in the full model, but these two factors were eliminated in the stepAIC procedure. We now report the model outputs and AIC values in the Supplementary material.

We have added an explanation to the text to clarify our strategy in the methods section under the “statistical analysis” (l 652-657), as well as in the results section (l 265 to 281) to provide more details on the selection procedure for our variables. In addition, we added a supplementary table (S4) with the outputs of the model and AIC values in the selection procedure.

We also modified the discussion (l396 to 411) since we can now eliminate the assumption related to the effect of chlorophyll-a on GD in the case of marine species (“In our study, the proxy for productivity (chlorophyll) was not retained in the final model. In addition, chlorophyll was only weakly correlated to temperature ($r = -0.10$, $p = 0.018$). Therefore, at this global scale, there is no support for the “energy-richness hypothesis”).

Result 4 seems interesting to me. However, It would deserve some improved analysis/discussion. In fact, if, as speculated by the authors, the higher GD among freshwater fish depends on the higher structural complexity of freshwater environment, we might expect a marked dependence on the geographic scale, with freshwater fish expected to harbour lower diversity within each water body but higher diversity within each cell of an appropriately scaled grid. It seems to me essentially similar to measuring alpha and beta diversity in ecology. I wonder if a more detailed analysis would not be possible using the same databases and some more geographic data (e.g., watersheds and river basins or exploring different grid sizes?). Similarly, the interpretation given for the differences among continents (that, e.g. South America hosts large river basins with high structural complexity), could have been straightforwardly tested by assigning cells to river basins and accounting for their size.

We thank the reviewer for this suggestion. We tested additional geographic variables in the model for freshwater species (upstream slope: mean, max, range, average (degree x100) and flow accumulation (count of number of upstream catchment grid cells). The variables were available at the following link: <http://www.earthenv.org/streams> at a resolution of about 1km. We also tested basin area as a proxy of basin size. Variables were averaged to follow our 200km grid resolution. We did not use soil or habitat variables as we do not think these are useful at this global spatial scale. We followed the procedure described in the model section to test the six additional explanatory variables (VIF procedure and step AIC after adding the autocorrelation variable). We detected that slope range significantly influences genetic diversity per cell (see figure below).

For marine species, we tested an additional variable that is distance to offshore. This can account for deeper water ocean. This variable does not significantly influence genetic diversity. We updated the final models with those results.

All in all, I suggest that the study may be improved by correcting a few statistical inaccuracies and, importantly, by attempting some more careful analysis/discussion of relevant geographic variables (e.g. size of river basins, river network complexity, actual distance among sampled conspecific individuals within cells, marine productivity). **Provided that, I think it has the potential to be published in Nature Communications.** We thank the reviewer for suggesting additional variables that can help to better describe the patterns. In the revised version, we have improved the statistical analyses when feasible. We have added careful analysis and discussion of relevant variables when possible (chlorophyll-a, elevation, slope and flow accumulation for freshwater species, distance to offshore for marine species).

I have marked more detailed comments (including some relatively important remarks on statistical methods) in a commented manuscript file.

We have answered these comments in the commented manuscript. See the file enclosed.

In the methods, it is not described how this barplot (Fig. 1b) was created. Is it a simple (weighted?) average across cells?

We have now added this explanation to the methods section 1 641-644: “For marine and freshwater species, genetic diversity per site was aggregated by latitude bands of 10°. We then plotted the genetic diversity per band of latitude using the R function barplot. The confidence interval for genetic diversity by latitude band was reported in the plot representing the variability of genetic diversity at latitudinal bands amongst sites (Fig. 1)”.

References

Albouy C, Guilhaumon F, Leprieur F, *et al.* (2013) Projected climate change and the changing biogeography of coastal Mediterranean fishes. *Journal Of Biogeography* **40**, 534-547.

- April J, Hanner RH, Mayden RL, Bernatchez L (2013) Metabolic Rate and Climatic Fluctuations Shape Continental Wide Pattern of Genetic Divergence and Biodiversity in Fishes. *Plos One* **8**, e70296.
- Bohonak AJ, Vandergast AG (2011) The value of DNA sequence data for studying landscape genetics. *Molecular Ecology* **20**, 2477-2479.
- Cheung W, Lam V, Sarmiento J, *et al.* (2009) *Projecting Global Marine Biodiversity Impacts under Climate Change Scenarios*.
- Cheung W, Pauly D, L. Sarmiento J (2013) *How to make progress in projecting climate change impacts*.
- Ellegren H, Galtier N (2016) Determinants of genetic diversity. *Nature Reviews Genetics* **17**, 422-433.
- Galtier N, Jobson Richard W, Nabholz B, Glémin S, Blier Pierre U (2009) Mitochondrial whims: metabolic rate, longevity and the rate of molecular evolution. *Biology Letters* **5**, 413-416.
- Lasram FB, Guilhaumon F, Albouy C, *et al.* (2010) The Mediterranean Sea as a 'cul-de-sac' for endemic fishes facing climate change. *Global Change Biology* **16**, 3233-3245.
- Saupe E, Hendricks J, Peterson A, S. Lieberman B (2014) *Climate change and marine molluscs of the western North Atlantic: Future prospects and perils*.
- Wang IJ (2010) Recognizing the temporal distinctions between landscape genetics and phylogeography. *Molecular Ecology* **19**, 2605-2608.

Reviewers' Comments:

Reviewer #3:

Remarks to the Author:

Compared to the first submission, the manuscript has certainly improved. I am largely satisfied with the author's reply to my statistical concerns, and I am glad that the authors made use of some of my suggestions and those of the other referees.

However, my opinion is that some lack of structural clarity and methodological detail make the manuscript not ready yet for publication in a high-profile journal such as Nature Communications.

I remark that I regard this as a valuable study which does deserve publication, but I urge the authors to make a further effort to provide potential readers with a concise, clear and consequential exposition of their work and its significance for macroecological theory. I hope my comments may help a bit.

More in detail, I still find that the conceptual structure of the manuscript is not completely linear. The hypotheses and predictions are not expressed in a consistent and clear way throughout the manuscript, and appear somewhat conflated.

For example, one important aim of the study consists of testing for spatial congruence between genetic diversity and species richness. However, it is still not possible to understand the significance of this congruence (or lack thereof) in the framework of ecological theory.

At L61-70 the "evolutionary speed", "colonization" and "energy-richness" hypotheses are described as "explaining spatial congruence between intra- and inter-specific levels of diversity"). Then, at L109-111, the formulation is rather different and confusing.

It may even seem like the "evolutionary speed" hypothesis would rather predict a decoupling of genetic diversity and species diversity. In fact, it is (maybe) contrasted to a generic H2 that "genetic diversity is positively correlated with species diversity as they are expected to be regulated by similar processes" (not very useful, indeed). This is inconsistent and prevents understanding of the significance of the study.

Interestingly, the "colonization" hypothesis is never mentioned anymore in the manuscript. Probably this is because it predicts the same as the "evolutionary speed" (higher diversity in the tropics), but the reader should be informed about its fate.

Moreover, at L111-113 it is stated that "habitat complexity and fragmentation [...] promotes genetic divergence (H3)". Does it also promote species diversity? (I would say it does).

To summarize, my general understanding of the background is that:

- "evolutionary speed", "colonization", "energy-richness" and (spatial) habitat complexity (see NOTE 1 below) may ALL foster diversity, BOTH at the gene (genetic diversity, GD) and species (species richness, SR) level. [No hypothesized mechanism is introduced that would predict decoupling of GD and SR, therefore, the significance of testing for this congruence is not very clear.]

- "regional" effects not explained by environmental variables may represent a layer of "structured" spatial autocorrelation of residuals, reflecting the anisotropy of the geographic space. To my understanding, they are just this.

The following predictions can be drawn from these premises:

1) Both the "evolutionary speed" and "colonization" hypothesis predict positive correlation of GD (and SR) with temperature. Unfortunately, there seems to be no way to disentangle the two, but that's it.

2) the "energy-richness" hypothesis predicts that chlorophyll (a proxy for productivity, and hence "energy richness") positively correlates with marine GD (and SR). As marine chlorophyll is weakly (and NEGATIVELY!) correlated with temperature, therefore, the "energy-richness" hypothesis predicts different patterns than the "evolutionary speed" and "colonization" hypotheses.

3) as (a certain kind of, see NOTE 1 below) habitat complexity is higher in freshwater than in marine environments, the "habitat complexity" hypothesis predicts that GD will be higher in freshwater fish. One might also think of some measure of the complexity of a river system and test the "habitat complexity" hypothesis within freshwater fish.

[I suggested that the authors investigate the effect of basin area because larger river basins may contain larger (somewhat) connected populations of each species than smaller basins. Therefore, I meant it as a test for the effect of spatial structure on population size (rather than population subdivision). I am a bit disappointed that, while this predictor was introduced in the analysis, its significance was not explained. I think that the significance of EACH predictor should be explained, and negative results should be discussed just as much as "significant" correlations. As far as I see it, this clearly marks the difference between "fishing expeditions" and hypothesis-driven science. See also my comment below on "slope range".]

The study found that:

1) Both GD and SR positively correlate with temperature, supporting "evolutionary speed" and "colonization" hypothesis.

2) Chlorophyll concentration was not a significant predictor of GD (nor SR?). No support for "energy richness".

3) Marine and freshwater fish have (on average) similar SR, but GD is higher in freshwater fish. It is not explained what the expectations about SR in marine vs. freshwater environments should be, however, this finding supports the "habitat complexity" hypothesis.

4) "Slope range" is (strongly) negatively correlated with freshwater GD. There is **NO** discussion whatsoever about what it means! We only know that "freshwater genetic diversity was mainly associated with geographic factors such as the regions and slope range". This is THE MOST IMPORTANT predictor of freshwater GD and, in fact, the reader is not even really allowed to understand what this variable actually is (see NOTE 2 below). I found this disconcerting.

All in all, it took me a considerable effort to derive this (subjective and hopefully correct) linearization of the study. I think it is very important that the average reader may follow a clear and consequential path from background hypothesis to predictions and back through the results, which the present manuscript is not yet delivering.

I marked a few detailed observations in an annotated version of the manuscript.

NOTE 1: Habitat complexity can mean many things. For example, a deep ocean has high complexity in that temperature, light, nutrients, pressure, etc. vary with depth, but low complexity in that it forms a single undivided body of water. In contrast, a river system may (not necessarily) have lower

complexity in terms of temp, light, etc, but will be highly complex in that it is formed by a composite network of watercourses (for simplicity, let us call this kind of complexity "spatial complexity"). Each species is typically adapted to live only in certain conditions of temperature, light, pressure, etc.. Therefore, "spatial complexity" promotes GD much more than it promotes SR, because allows for several (more or less) isolated populations of the *same species*, while kinds of habitat complexity may allow for the coexistence (in the same 2D cell) of *different species*.

I think this may explain why SR is similar in freshwater and marine fish, while GD is higher in freshwater fish.

I don't know if this may have anything to do with the *negative* correlation of "slope range" and GD in freshwater fish, but it may be worth thinking about it.

NOTE 2: The main text lacks any detail. Supplementary Table 3 lists "Slope and flow accumulation" as a predictor. The single other mention of "slope and flow accumulation" is in the caption of Supplementary Table 5, while, everywhere else, I only found "slope range". Interestingly, as far as I could see, the web address "<http://www.earthenv.org/streams>" (cited in Supplementary Table 3 and consulted by me on 16 Jul 2019), does not list any variable called "slope and flow accumulation" nor "slope". I could only find "Upstream slope (min, max, range, avg)" and "Stream length and flow accumulation". It seems they are two rather different things. In any case, it is not possible to know what went into the analysis. I suspect this may be a significant finding from this study. I think readers should know what it is.

Reviewer #4:

Remarks to the Author:

I believe that the authors adequately addressed most the reviewers' comments and suggestions, and the revised manuscript is significantly improved. Below I provide my assessment on how the authors responded to the reviewers' main concerns, with some recommendations that could further improve the manuscript.

Reviewer #1:

One main concern of reviewer 1 was the effect of data availability (amount of genetic data and taxonomic coverage) per cell on the reported patterns. The authors have largely addressed these concerns and the model explanatory power has increased considerably (from 16% for all the cells up to 34% of the variance explained for the filtered data sets). However, regarding the effect of taxonomic coverage within each grid cell, the authors used the absolute number of sampled species instead of the percentage of sampled species relative to the actual species diversity in each cell (which is the definition of taxonomic coverage for each cell). I suggest two alternative solutions, with the first one being slightly more demanding but in line with the reviewer's suggestion:

1. Explore the effect of varying the actual taxonomic coverage per cell, e.g. include only cells with a minimum taxonomic coverage of 2%, 5% etc. (plus a minimum number of sequences per cell to avoid discarding data rich cells). The main rationale behind this suggestion is that cells with low taxonomic coverage (e.g. a cell with 8 sampled species when the actual species diversity is 200 represents only 4% of the actual diversity, while a grid cell with 4 species sampled and a species diversity of 50 would represent 8% of the actual diversity. Patterns are not expected to change drastically, but this one is a more meaningful filtering criterium.
2. Rewrite lines 258 – 264 to better reflect what the authors actually did (i.e. applied a minimum absolute number of sampled species and not a minimum taxonomic coverage). This is the easiest solution and will not confuse the readers.

Most importantly, and related to comment above, the authors do not report correlations between the

filtered datasets and species diversity (i.e. richness), which is one of the main themes of the manuscript (Fig. 1; first sentence of the discussion and main conclusion at the end of the discussion). Since the model explanatory power is much higher on the filtered datasets, I would expect the same pattern for species richness. This would further reinforce the role of the "evolutionary speed hypothesis" that links micro and macroevolution through higher energy availability. Additionally, it will further reinforce the conservation aspects of the manuscript by revealing whether there is actually a mismatching spatial signal between species and genetic diversity.

Regarding the third major comment of reviewer 1 on an "extended description of protocols and decisions behind the alignment of sequences", the authors responded only within the response letter but not in the main text. I suggest to add this info (e.g. minimum sequence overlap) also in the main text, unless the editors find the reference to Miraldo et al. as adequate source of methods.

In their fourth major comment, the reviewer suggested a discussion regarding the relevance of sea-surface temperature for fishes inhabiting deeper waters. The authors opted not to add this multi-argument discussion in the paper. I understand that adding it may increase the length of the paper, but at least one sentence describing the high correlation between surface and deeper sea temperature would be valuable for the general reader.

Reviewer #2:

The authors sufficiently address all of the comments of the second reviewer, either in the response letter or in the main manuscript.

Reviewer #3:

The authors sufficiently addressed most the reviewer's comments. I have one suggestion regarding the reviewer's comment on the latitudinal band gradient.

"In the methods, it is not described how this barplot (Fig. 1b) was created. Is it a simple (weighted?) average across cells?"

The authors now describe the methodology behind the creation of the barplots in Fig. 1. However, since the grid cells arbitrarily split the distribution of intraspecific sequences into smaller sites, and since the utilized datasets suffer from sampling issues and noise, wouldn't it be more meaningful to intersect sequences with each band (polygon), with confidence intervals representing the GD variability across species? This suggestion/approach has been criticised for not considering geographic distances between intraspecific sequences across latitudinal bands (Gratton et al. 2017, Ecology Letters; ref 27 in the main text). However, what the authors are currently doing is averaging over both "good" and "bad" cells, so this is expected to create a lot of noise in the patterns observed. One possible solution is to average over data-rich cells only. Nevertheless, I still believe that the Miraldo et al. approach makes more sense, especially when contrasting GD patterns with species richness.

Minor comments:

line 99: BOLD is first referenced without full description, i.e. Barcode of Life Database.

Detailed answers to Reviewers' comments:

Reviewer #3 (Remarks to the Author):

Compared to the first submission, the manuscript has certainly improved. I am largely satisfied with the author's reply to my statistical concerns, and I am glad that the authors made use of some of my suggestions and those of the other referees. However, my opinion is that some lack of structural clarity and methodological detail make the manuscript not ready yet for publication in a high-profile journal such as Nature Communications. I remark that I regard this as a valuable study which does deserve publication, but I urge the authors to make a further effort to provide potential readers with a concise, clear and consequential exposition of their work and its significance for macroecological theory. I hope my comments may help a bit.

We followed these recommendations and made substantial efforts to provide a clearer and more logical consequential exposition of our work, as well as to add missing methodological details.

More in detail, I still find that the conceptual structure of the manuscript is not completely linear. The hypotheses and predictions are not expressed in a consistent and clear way throughout the manuscript, and appear somewhat conflated. For example, one important aim of the study consists of testing for spatial congruence between genetic diversity and species richness. However, it is still not possible to understand the significance of this congruence (or lack thereof) in the framework of ecological theory.

We agree with the reviewer that the test for spatial congruence between genetic diversity and species richness was not well justified and weakly interpreted in the previous version of the manuscript. Genetic diversity is expected to follow clear biogeographic patterns, but there is currently only a limited comprehension of large-scale genetic biogeography. Those patterns might be congruent with those of species diversity as a result of multiple processes occurring along a micro- to macroevolution continuum. This was suggested by local-scale studies demonstrating a correlation between spatial genetic and species diversity (eg : Bertin *et al.* 2017; Lamy *et al.* 2017; Vellend 2005).

From this background, our paper aims (i) to show (or not) non-random geographic distribution of genetic diversity, (ii) then to detect congruence between genetic and species diversity, if any, and (iii) to reveal the determinants genetic diversity patterns given the knowledge of the processes driving species richness patterns. To clarify this message, we better explained the continuum between micro and macro evolutionary processes but, overall, we added alternative assumptions and we inverted the order of the assumptions in the introduction (see below the new hypotheses). Finally, we modified the discussion accordingly.

The new paragraph reads as follow (lines 116-129):

“We hypothesized that genetic diversity should display a non-random spatial distribution at the global scale (H1), and that freshwater habitats have higher genetic diversity than the marine realm given the higher habitat complexity (e.g. composite network of watercourses) and fragmentation, which both promote genetic divergence (H2). Then, the micro-macro continuum hypothesis predicts that similar processes may act on both species and genetic

diversity, and in consequence we should detect a positive correlation between them (H3). In more details, the evolutionary speed hypothesis and the colonization hypothesis predict that temperature should be positively correlated to fish genetic diversity (H3.1). The energy-richness hypothesis predicts that productivity positively correlates with genetic diversity (H3.2). Finally, diversity might vary across regions because of underlying region-specific environmental parameters (H3.3). For each of these hypotheses, the null expectation corresponds to an absence of spatial structure and no effect of factors (i.e. no spatial pattern of genetic diversity, no congruence between genetic and species diversity, no influence of temperature, productivity, and geographic region).”

We think that those changes will help to “linearize” the structure of the manuscript as recommended by the reviewer and will make the reading easier and more logical.

At L61-70 the "evolutionary speed", "colonization" and "energy-richness" hypotheses are described as "explaining spatial congruence between intra- and inter-specific levels of diversity". Then, at L109-111, the formulation is rather different and confusing. It may even seem like the "evolutionary speed" hypothesis would rather predict a decoupling of genetic diversity and species diversity. In fact, it is (maybe) contrasted to a generic H2 that "genetic diversity is positively correlated with species diversity as they are expected to be regulated by similar processes" (not very useful, indeed). This is inconsistent and prevents understanding of the significance of the study.

We changed the formulation of our hypothesis in order to make this point clearer. See the new formulation in the previous comment and lines 116-129 in the paper.

Interestingly, the "colonization" hypothesis is never mentioned anymore in the manuscript. Probably this is because it predicts the same as the "evolutionary speed" (higher diversity in the tropics), but the reader should be informed about its fate.

The colonization hypothesis is related to the instability and demographic hypotheses. We agree that it was unclear in the previous version of the manuscript and we clarified this point in the introduction.

It now reads (lines 68-75):

“Second, a positive association between species, genetic diversity, and temperature is expected under the “colonization hypothesis” (or “stability hypothesis”). This hypothesis posits that unstable regions are associated with events causing local extinctions of individuals and/or species. These events are generally followed by stochastic recolonization generating bottlenecks, which may lower both species and genetic local diversity (Mittelbach *et al.* 2007). Typically, warmer areas in the tropics have suffered less variability over geological times, whereas cold areas were highly unstable generating species diversity clines along temperature gradients (Pellissier *et al.* 2014).”

Moreover, at L111-113 it is stated that "habitat complexity and fragmentation [...] promotes genetic divergence (H3)". Does it also promote species diversity? (I would say it does).

Indeed, it does. We rewrote the sentence in the introduction to add this information as well as to clarify the meaning of habitat complexity.

It now reads (lines 77-80):

“Finally, an alternative hypothesis states that areas with higher physical complexity should provide more different habitats and hence support higher species diversity (Grosberg *et al.* 2012; Kovalenko *et al.* 2012), but also more diverse population structures leading to higher genetic diversity (Bertin *et al.* 2017)

To summarize, my general understanding of the background is that:

- "evolutionary speed", "colonization", "energy-richness" and (spatial) habitat complexity (see NOTE 1 below) may ALL foster diversity, BOTH at the gene (genetic diversity, GD) and species (species richness, SR) level. [No hypothesized mechanism is introduced that would predict decoupling of GD and SR, therefore, the significance of testing for this congruence is not very clear.]

No mechanism is proposed in the literature to explain a "decoupling" between species and genetic diversity because the theory (Laroche *et al.* 2015; Vellend 2005) predicts a coupling.

However, we now introduced explanations for a weak congruence between both patterns when we discussed the limits of our approach (lines 319-325):

“The relationship between genetic and species diversity is weak. The processes underlying genetic diversity distribution might show differences to those underlying species diversity patterns in distinct habitats and geographic areas. Those differences can be explained by disparities in temporal and spatial scales or in responses to environmental changes of the parallel ecological and evolutionary processes (mutation vs speciation; genetic vs ecological drift; gene flow vs dispersal; selection vs environmental filter) (Fourtune *et al.* 2016).”

- "regional" effects not explained by environmental variables may represent a layer of "structured" spatial autocorrelation of residuals, reflecting the anisotropy of the geographic space. To my understanding, they are just this.

We agree with the reviewer and we removed any reference to the effects produced by other variables in the results and in the discussion.

The following predictions can be drawn from these premises:

- 1) Both the "evolutionary speed" and "colonization" hypothesis predict positive correlation of GD (and SR) with temperature. Unfortunately, there seems to be no way to disentangle the two, but that's it.
- 2) the "energy-richness" hypothesis predicts that chlorophyll (a proxy for productivity, and hence "energy richness") positively correlates with marine GD (and SR). As marine chlorophyll is weakly (and NEGATIVELY!) correlated with temperature, therefore, the "energy-richness" hypothesis predicts different patterns than the "evolutionary speed" and "colonization" hypotheses.
- 3) as (a certain kind of, see NOTE 1 below) habitat complexity is higher in freshwater than in marine environments, the "habitat complexity" hypothesis predicts that GD will be higher in freshwater fish.

We thank the reviewer for this nice synthesis of our results. We follow reviewer' suggestions and we integrated all the predictions in the introduction. This greatly helps to clarify the main assumption of our paper. See (lines 116-129) and answer to the first comment. We also added explanations in the introduction.

One might also think of some measure of the complexity of a river system and test the "habitat complexity" hypothesis within freshwater fish.

[I suggested that the authors investigate the effect of basin area because larger river basins may contain larger (somewhat) connected populations of each species than smaller basins. Therefore, I meant it as a test for the effect of spatial structure on population size (rather than population subdivision). I am a bit disappointed that, while this predictor was introduced in the analysis, its significance was not explained. I think that the significance of EACH predictor should be explained, and negative results should be discussed just as much as "significant" correlations. As far as I see it, this clearly marks the difference between "fishing expeditions" and hypothesis-driven science. See also my comment below on "slope range".]

We now discussed non-significant predictors and negative results in our interpretation (line 259-270):

“The strong negative relationship between the average slope of river basins and genetic diversity is theoretically expected given that steeper rivers are defined by harsher and less stable habitats (hence lower population sizes and lower genetic diversity) and, that steeper rivers have been less prone to intense re-colonization after the last glaciation, which also tends to decrease genetic diversity. The region effect as well as the spatial autocorrelation factor appear as good surrogates of habitat complexity at global scale. Surprisingly, some factors like total basin area were not significantly associated to genetic diversity. Indeed, theoretically, basin area should correlate positively with regional genetic diversity since higher effective population sizes should be supported in larger river basins. This later result and the fact that genetic diversity was negatively associated to the river slope suggest that colonisation processes might be more important than genetic drift in explaining patterns of genetic diversity in freshwater systems, as previously suggested locally”

Additionally, chlorophyll was not selected as significant in the model for marine species. Since chlorophyll was only weakly and negatively correlated to temperature ($r = -0.10$, $P = 0.018$), our results suggest that temperature-dependant productivity is unlikely to modulate diversity through population size of marine species. It reads (lines 275-278):

“Chlorophyll a, as a surrogate for productivity, was not retained in the final model and was weakly and negatively correlated to temperature ($r = -0.10$, $P = 0.018$). This result suggests that temperature-dependent productivity is unlikely to modulate genetic diversity through population sizes in marine species”

The study found that:

- 1) Both GD and SR positively correlate with temperature, supporting "evolutionary speed" and "colonization" hypothesis.**
- 2) Chlorophyll concentration was not a significant predictor of GD (nor SR?). No support for "energy richness".**
- 3) Marine and freshwater fish have (on average) similar SR, but GD is higher in freshwater fish. It is not explained what the expectations about SR in marine vs. freshwater environments should be, however, this finding supports the "habitat complexity" hypothesis.**
- 4) "Slope range" is (strongly) negatively correlated with freshwater GD. There is *NO* discussion whatsoever about what it means! We only know that "freshwater genetic diversity was mainly associated with geographic factors such as the regions and slope range". This is THE MOST IMPORTANT predictor of freshwater GD and, in fact, the**

reader is not even really allowed to understand what this variable actually is (see NOTE 2 below). I found this disconcerting.

All in all, it took me a considerable effort to derive this (subjective and hopefully correct) linearization of the study. I think it is very important that the average reader may follow a clear and consequential path from background hypothesis to predictions and back through the results, which the present manuscript is not yet delivering.

We hope that the new version of the manuscript delivers a clearer message. We deeply changed both the introduction to clarify the predictions and the discussion to clarify the interpretations.

Related to the point 4 (negative relation), you are completely right that the range of slope explained a large part of the variance in our model. Yet, we would rather expect a positive relationship between slope range and genetic diversity, since a higher range would indicate a higher habitat complexity and hence a higher genetic diversity. We therefore went deeper into this analysis and we reconsidered the variables to be included in the model. First, we removed from the initial model the minimum and maximum slopes (average and absolute) that make little sense ecologically. So, we only considered the average slope and the range of slope in the basin. The problem is that (because of the Taylor's law) both are highly correlated (positively). In the new model we only kept the mean slope that is the one selected by AIC when both are in the initial model and for which we have better theoretical and biological interpretation (Paz-Vinas & Blanchet 2015). We found a negative relationship between mean slope and genetic diversity which is theoretically expected given that (i) steeper rivers are defined by harsher and less stable habitats (hence lower population sizes and lower genetic diversity) and (ii) steeper rivers have been less prone to intense re-colonization after the last glaciation, which also tends to decrease genetic diversity. We now discuss further this finding (lines 275-264; see above comments).

I marked a few detailed observations in an annotated version of the manuscript.

We have accounted for all observations of the annotated version. See answers directly in the manuscript.

NOTE 1: Habitat complexity can mean many things. For example, a deep ocean has high complexity in that temperature, light, nutrients, pressure, etc. vary with depth, but low complexity in that it forms a single undivided body of water. In contrast, a river system may (not necessarily) have lower complexity in terms of temp, light, etc, but will be highly complex in that it is formed by a composite network of watercourses (for simplicity, let us call this kind of complexity "spatial complexity").

We have now clarified what we call habitat complexity. In our study we refer to the physical component of the habitat: i.e. spatial complexity since temperature for example is already accounted as an explicit variable.

In the introduction we added (lines 116-119): “We hypothesized that genetic diversity should display a non-random spatial distribution at the global scale (H1), and that freshwater habitats have higher genetic diversity than the marine realm given the higher habitat spatial complexity (e.g. composite network of watercourses) and fragmentation, which both promote genetic divergence (H2)”

Each species is typically adapted to live only in certain conditions of temperature, light, pressure, etc.. Therefore, "spatial complexity" promotes GD much more than it promotes SR, because allows for several (more or less) isolated populations of the *same species*, while kinds of habitat complexity may allow for the coexistence (in the same 2D cell) of *different species*.

I think this may explain why SR is similar in freshwater and marine fish, while GD is higher in freshwater fish.

We agree with the reviewer and we added this argument in the discussion lines 249-252:
"This habitat spatial complexity may promote genetic diversity much more than it promotes species diversity, because it can favour the persistence of isolated populations within species in a very restricted area, but not the coexistence of a large number of species in competition within the same water segment."

I don't know if this may have anything to do with the *negative* correlation of "slope range" and GD in freshwater fish, but it may be worth thinking about it.

We now discuss the interpretation of the slope effect in more details. See previous answer to the related comment.

NOTE 2: The main text lacks any detail. Supplementary Table 3 lists "Slope and flow accumulation" as a predictor. The single other mention of "slope and flow accumulation" is in the caption of Supplementary Table 5, while, everywhere else, I only found "slope range". Interestingly, as far as I could see, the web address "<http://www.earthenv.org/streams>" (cited in Supplementary Table 3 and consulted by me on 16 Jul 2019), does not list any variable called "slope and flow accumulation" nor "slope". I could only find "Upstream slope (min, max, range, avg)" and "Stream length and flow accumulation". It seems they are two rather different things. In any case, it is not possible to know what went into the analysis. I suspect this may be a significant finding from this study. I think readers should know what it is.

We added more details in the main text (method and results), and the technical description of the variables are given in the supplementary Table 3. The variables that we used are referred as topography category and upstream slope (min, max, range, avg, min and max absolute), and flow accumulation. We selected from this database only the variables that are potential drivers of genetic diversity patterns and that can be interpreted ecologically.

See (lines 440-445) of the manuscript:

"The geographical factors included geographic coordinates in degree, regions, bathymetry and distance to offshore accounting for deeper water ocean for marine species only, and elevation, basin area, slopes (average and range) and flow accumulation as a surrogate of watershed size for freshwater species only (Supplementary Table 3). For slope-related variables we chose to keep in our analyses only values with biological interpretation"

And in the legend of the supplementary Table 3:

"Only for freshwater species. Values were obtained from Domisch et al (2015) and are available from a grid of resolution 1km. For our analysis, we averaged all the 1km pixels in

each 200 km cell of our grid y using the "extract" function from the package raster. Flow accumulation is the amount of upstream area draining into each cell and is measured in count of grid cells. It could be considered as a surrogate of watershed size. Slope Units: ($[^\circ] * 100$). Slopes values (average and range in our analysis) are estimated from the upstream slope of each cell of the 1km grid."

Reviewer #4 (Remarks to the Author):

I believe that the authors adequately addressed most the reviewers' comments and suggestions, and the revised manuscript is significantly improved. Below I provide my assessment on how the authors responded to the reviewers' main concerns, with some recommendations that could further improve the manuscript.

We are happy to nearly match to previous reviewer's suggestions. We carefully considered your additional recommendations to improve our manuscript. See below answers.

Reviewer #1:

One main concern of reviewer 1 was the effect of data availability (amount of genetic data and taxonomic coverage) per cell on the reported patterns. The authors have largely addressed these concerns and the model explanatory power has increased considerably (from 16% for all the cells up to 34% of the variance explained for the filtered data sets). However, regarding the effect of taxonomic coverage within each grid cell, the authors used the absolute number of sampled species instead of the percentage of sampled species relative to the actual species diversity in each cell (which is the definition of taxonomic coverage for each cell). I suggest two alternative solutions, with the first one being slightly more demanding but in line with the reviewer's suggestion:

1. Explore the effect of varying the actual taxonomic coverage per cell, e.g. include only cells with a minimum taxonomic coverage of 2%, 5% etc. (plus a minimum number of sequences per cell to avoid discarding data rich cells). The main rationale behind this suggestion is that cells with low taxonomic coverage (e.g. a cell with 8 sampled species when the actual species diversity is 200 represents only 4% of the actual diversity, while a grid cell with 4 species sampled and a species diversity of 50 would represent 8% of the actual diversity. Patterns are not expected to change drastically, but this one is a more meaningful filtering criterium.

We followed both recommendations and we added in the revised version a sensitivity analysis to taxonomic coverage per cell on patterns and model outputs (Supplementary Tables 6 and 7) as well as a map of the taxonomic coverage per cell (Supplementary Figure 7).

-Methods is described lines (483-488):

"We finally investigated the influence of more stringent thresholds for absolute species or sequences number and taxonomic coverage to estimate genetic diversity per cell and thus test the robustness of our findings to some arbitrary choices (number of species and sequences per cell and taxonomic coverage per cell). We choose thresholds that allowed to select around one-third and two-third of the "best" grid cells when possible (Supplementary Tables 6,7). The taxonomic coverage used for this analysis is mapped in the Supplementary Figure 7."

-Results are reported (lines 213-221):

“For marine species, removing cells with less than 4 sequences (at least 2 species), or than 8 species (at least 2 sequences), or filtering for a taxonomic coverage of 5% when estimating genetic diversity had marginal effect on model outputs (Supplementary Table 7). The explanatory power (adjusted R²) of the models always increased except when only less than 34% of cells were retained after filtering for cells with at least two species and with more than four sequences or with a taxonomic coverage per cell of 5% in each cell. For freshwater species, we also observed unchanged results with more stringent filters, and the main factor (region) was significant in all cases (Supplementary Table 7). The explanatory power (R²) also increased in all cases except for a taxonomic coverage per cell of 5%.”

See new supplementary Table 7. As expected, the patterns do not change drastically.

2. Rewrite lines 258 – 264 to better reflect what the authors actually did (i.e. applied a minimum absolute number of sampled species and not a minimum taxonomic coverage). This is the easiest solution and will not confuse the readers.

We also clarified what we did: see the answer to the point 1 in the method section, line 483 we added “absolute” species and sequence number.

Most importantly, and related to comment above, the authors do not report correlations between the filtered datasets and species diversity (i.e. richness), which is one of the main themes of the manuscript (Fig. 1; first sentence of the discussion and main conclusion at the end of the discussion). Since the model explanatory power is much higher on the filtered datasets, I would expect the same pattern for species richness. This would further reinforce the role of the “evolutionary speed hypothesis” that links micro and macroevolution through higher energy availability. Additionally, it will further reinforce the conservation aspects of the manuscript by revealing whether there is actually a mismatching spatial signal between species and genetic diversity.

The reviewer is correct. We now added a sensitivity analysis to test the robustness of the correlation between genetic diversity and species diversity. The method is described lines (483-488) (see above). We found that (lines 208-213):

“Filtering for taxonomic coverage per cell when estimating genetic diversity influenced the correlation between genetic and species diversity; the modified t-test of the spatial dependence is only significant when the taxonomic coverage per cell is $\leq 1\%$ for marine species and 0% for freshwater species (Supplementary Table 6). Conversely, less stringent filters on the number of sequences or species, used as surrogates for sampling effect, changed only slightly the results for this pattern (Supplementary Table 6).”

We reported the full results in the Supplementary Table 6.

Regarding the third major comment of reviewer 1 on an “extended description of protocols and decisions behind the alignment of sequences”, the authors responded only within the response letter but not in the main text. I suggest to add this info (e.g. minimum sequence overlap) also in the main text, unless the editors find the reference to Miraldo et al. as adequate source of methods.

We have included the details about sequence alignments the in the main text lines 381-384:

“The alignments were checked manually using the software ugene. Additionally, only pairwise alignments with overlap greater than 50% were kept to calculate genetic diversity.”

In their fourth major comment, the reviewer suggested a discussion regarding the relevance of sea-surface temperature for fishes inhabiting deeper waters. The authors opted not to add this multi-argument discussion in the paper. I understand that adding it may increase the length of the paper, but at least one sentence describing the high correlation between surface and deeper sea temperature would be valuable for the general reader.

As recommended by the reviewer, we added a short discussion about the relevance of sea-surface temperature for fish inhabiting deeper waters. Lines 331-339 it reads:

“We used sea-surface temperature to explain patterns of genetic diversity for fishes while some inhabit deeper waters. They represent about 40% of fish species (bathymersal, bathypelagic, and demersal species) potentially not living close to the surface among the 3,815 marine species considered in our study. Yet, sea surface temperature is strongly correlated with sea bottom temperature at the scale of continental shelf (0-200m)⁵⁶. In addition, in our case, if sea surface temperature was a poor predictor for those species living in deeper seawaters, we should have detected an effect of the distance to offshore in the model. However, this factor was not shown to significantly influence mean genetic diversity per cell.”

Reviewer #2:

The authors sufficiently address all of the comments of the second reviewer, either in the response letter or in the main manuscript.

We are happy to have satisfied your requests.

Reviewer #3:

The authors sufficiently addressed most the reviewer’s comments. I have one suggestion regarding the reviewer’s comment on the latitudinal band gradient.

“In the methods, it is not described how this barplot (Fig. 1b) was created. Is it a simple (weighted?) average across cells?”

The authors now describe the methodology behind the creation of the barplots in Fig. 1. However, since the grid cells arbitrarily split the distribution of intraspecific sequences into smaller sites, and since the utilized datasets suffer from sampling issues and noise, wouldn’t it be more meaningful to intersect sequences with each band (polygon), with confidence intervals representing the GD variability across species? This suggestion/approach has been criticised for not considering geographic distances between intraspecific sequences across latitudinal bands (Gratton et al. 2017, Ecology Letters; ref 27 in the main text). However, what the authors are currently doing is averaging over both “good” and “bad” cells, so this is expected to create a lot of noise in the patterns observed. One possible solution is to average over data-rich cells only. Nevertheless, I still believe that the Miraldo et al. approach makes more sense, especially when contrasting GD patterns with species richness.

We confirm that this barplot represents the averaged genetic diversity across cells within a given latitudinal band of 10°: we added this information in the legend of Figure 1.

We also followed the reviewer's recommendation to provide a figure similar to that of Miraldo showing mean genetic diversity by latitude band. In practice we calculated mean intraspecific genetic diversity (across species) in each 10° latitudinal band. To estimate confidence interval of this mean, we did 1,000 bootstrap replicates (removing in each band one species each time). We then estimated and represented standard deviation in each latitudinal band from the 1,000 bootstrapped replications. The figures are given below for marine and freshwater fishes both using no filter on species and sequences and removing species with null genetic diversity or/and less than 3 individuals.

Anyway, it will never be possible to obtain exactly the same data that we used in our actual representation of the genetic diversity averaged by cell (Figure 1, b and d) since not the same filters can be applied. We think that for the clarity our manuscript, it is better to keep the figure that was produced using the same data that the one used for our map and models, i.e. genetic diversity averaged across cells (instead averaged across species).

Figure. Genetic diversity averaged across species by latitudinal band of 10° (a) for Marine species (b) For Freshwater species. Standard deviation was estimated from 1000 bootstrapped replications and represented on each boxplot bar. We filtered out species with no genetic diversity or/and less than 3 individuals.

Comparing the two strategies, we can check that the patterns are conserved for both species. The peak is slightly different: north latitudinal bands at 20-40° instead of 10-20°.

Concerning the suggestion to keep only the good cells in our current version of the figure, yet the filter used to produce data for the model on species and sequences already selects cells with usable data. So, we did not investigate further for this new figure.

Minor comments:

line 99: **BOLD** is first referenced without full description, i.e. Barcode of Life Database.

We changed it.

References

- Bertin A, Gouin N, Baumel A, *et al.* (2017) Genetic variation of loci potentially under selection confounds species-genetic diversity correlations in a fragmented habitat. *Molecular Ecology* **26**, 431-443.
- Fourtune L, Paz-Vinas I, Loot G, Prunier JG, Blanchet S (2016) Lessons from the fish: a multi-species analysis reveals common processes underlying similar species-genetic diversity correlations. *Freshwater Biology* **61**, 1830-1845.
- Grosberg RK, Vermeij GJ, Wainwright PC (2012) Biodiversity in water and on land. *Current Biology* **22**, R900-R903.
- Kovalenko KE, Thomaz SM, Warfe DM (2012) Habitat complexity: approaches and future directions. *Hydrobiologia* **685**, 1-17.
- Lamy T, Laroche F, David P, Massol F, Jarne P (2017) The contribution of species-genetic diversity correlations to the understanding of community assembly rules. *Oikos* **126**, 759-771.
- Laroche F, Jarne P, Lamy T, David P, Massol F (2015) A Neutral Theory for Interpreting Correlations between Species and Genetic Diversity in Communities. *American Naturalist* **185**, 59-69.
- Mittelbach GG, Schemske DW, Cornell HV, *et al.* (2007) Evolution and the latitudinal diversity gradient: speciation, extinction and biogeography. *Ecology Letters* **10**, 315-331.
- Paz-Vinas I, Blanchet S (2015) Dendritic connectivity shapes spatial patterns of genetic diversity: a simulation-based study. *Journal Of Evolutionary Biology* **28**, 986-994.
- Pellissier L, Leprieur F, Parravicini V, *et al.* (2014) Quaternary coral reef refugia preserved fish diversity. *Science* **344**, 1016-1019.
- Vellend M (2005) Species diversity and genetic diversity: Parallel processes and correlated patterns. *American Naturalist* **166**, 199-215.

Reviewers' Comments:

Reviewer #3:

Remarks to the Author:

The new version of the manuscript is a further, substantial improvement, and I feel that the study is close to being ready for publication.

Nonetheless, I still have some relatively important observations, which follow.

Moreover, I have included a pretty large number of more or less important suggestions in a commented version of the manuscript. I hope that the authors will take it as a further constructive effort from my side, and will carefully consider my suggestions before final publication.

In particular, I warmly appreciate the new presentation of the main explanatory hypotheses for the geographic distribution of diversity (both at the specific and genetic level) at lines 61-81. I have the impression, though, that the listing of predictions at lines 117-130 is not fully consistent with it. In fact, I found it a bit confusing, and probably not really necessary, since all predictions are already rather clear from lines 61-81 (maybe, it could be added something for the 'physical complexity' hypothesis, which would link it more directly to the marine-freshwater contrast).

A disappointment was that, although the authors improved the discussion by commenting about all predictors, they did not yet fully explain, in the Introduction and/or Methods, the precise rationale for the choice of each of the predictors in their linear models. I think this should be further improved.

Some interpretations still look quite a bit outstretched.. for example, it is not explained why "the region effect as well as the spatial autocorrelation factor appear as good surrogates of habitat complexity". Also, it is important that the authors avoid sentences like "our results support the evolutionary speed hypothesis". There is no way to disentangle the "speed" and "colonization" hypotheses with this data. Therefore these results *do not* "support" either. They are consistent with both! This should be made absolutely clear to the readers.

Importantly, it is not explained how confidence intervals for model coefficients were obtained. As P-values are biased by model selection (Line 476), these CIs are the most important hint for the significance of predictors!

I suggest some restructuring to the Discussion, which is quite redundant and not very linear (see the commented manuscript for details).

Paolo Gratton

Reviewer #4:

Remarks to the Author:

I believe that the authors revised the manuscript appropriately by taking all reviewers' considerations into account. Below, I provide a few minor comments to be addressed before publication of the manuscript.

Abstract:

While the authors lay out a clear expectation (the positive relationship between GD and SR) in lines

26-27, they do not mention that they are actually testing for that in the next lines (lines 29-30), but they mention it only as a result (lines 31-32).

Results:

Please also mention in the methods the genetic marker (co1) that was used for the inferences

Lines 144-147: The authors mention that a peak of genetic diversity occurs in the North latitudinal band of 10-20 degrees latitude (Fig. 1 b,d). However, I do not see this peak in the figure. I see peaks in the South Latitudinal bands (-10 to -20 degrees). Please correct.

Line 150: What do the authors mean by "median genetic diversity per cell". I thought the genetic diversity was calculated as the mean across all sampled species per cell (395-396). Maybe they meant that they compared the median values of the two distributions (freshwater vs marine).

Line 179-178: Change "because detected" to "because it was identified as"

Line 192: I am missing a statement in the Methods on how the relative explained variance was calculated.

Discussion:

244-246: Is there a relevant reference that states that connectivity (i.e. gene flow) reduces genetic variation (i.e. number of mutations) at the nucleotide level? If so, please add it.

Lines 318-324: I suggest to move this part in the previous paragraph (Line 299) where the authors discuss the intra- interspecific diversity associations.

Figures:

Figure 3: "Determinants" instead of "Determinant" in title?

Supplementary Figure 4: Please add (a) and (b) letters in the subplots

Sincerely,
Spyros Theodoridis

Reviewers' comments:

Reviewer #3 (Remarks to the Author): to Paolo Gratton

The new version of the manuscript is a further, substantial improvement, and I feel that the study is close to being ready for publication.

Nonetheless, I still have some relatively important observations, which follow. Moreover, I have included a pretty large number of more or less important suggestions in a commented version of the manuscript. I hope that the authors will take as a further constructive effort from my side, and will carefully consider my suggestions before final publication.

We really appreciate the reviewer's effort to improve our MS. We integrated all the comments in the revision.

In particular, I warmly appreciate the new presentation of the main explanatory hypotheses for the geographic distribution of diversity (both at the specific and genetic level) at lines 61-81. I have the impression, though, that the listing of predictions at lines 117-130 is not fully consistent with it. In fact, I found it a bit confusing, and probably not really necessary, since all predictions are already rather clear from lines 61-81 (maybe, it could be added something for the 'physical complexity' hypothesis, which would link it more directly to the marine-freshwater contrast).

Glad to know that the reviewer appreciated the new presentation.

We added explanation for the physical complexity hypothesis lines 77 to 83 :

« Finally, the “physical complexity hypothesis” states that areas with higher habitat complexity should provide more ecological niches and hence support higher species diversity^{27, 28}, but also more spatially structured populations in a given area so a higher genetic diversity²⁹. This “physical complexity hypothesis” strongly depends upon the spatial grain and extent of the study area. For instance, at a large scale, a complex network of watercourses, typically characterizing freshwater habitats, should promote higher genetic (and species) diversity than more homogenous and continuous marine waters. »

We removed the predictions lines 117-130 and replaced them by a short summary of the hypotheses lines 115-117.

« We interpreted our results according to the micro-macro continuum concept and in the light of the evolutionary speed, colonisation, energy and habitat complexity hypotheses. »

A disappointment was that, although the authors improved the discussion by commenting about all predictors, they did not yet fully explain, in the Introduction and/or Methods, the precise rationale for the choice of each of the predictors in their linear models. I think this should be further improved.

We added some more rationale for the choice of our predictors in the method section lines 450 to 464:

“Basin area has the same value for all the cells from a given basin, whereas flow accumulation provides a local cell estimation of the watershed size, with upstream cells having lower values than downstream cells. Theoretically, basin area should correlate positively with genetic diversity since higher regional (basin-scale) effective population sizes

should be supported in larger river basins. An effect of the basin area reflects regional-scale processes and can be interpreted for instance in terms of past history (e.g. founder effects due to past colonisation) or connectivity at the scale of the basin. We are also expecting a positive correlation between flow accumulation and genetic diversity through processes acting at the local scale; for instance, higher flow accumulation suggests higher local effective populations sizes and hence higher genetic diversity, irrespective of the basin area. For slope-related variables, a negative relationship between the average slope of river basins and genetic diversity is theoretically expected given that steeper rivers are characterized by smaller and less stable hydrological conditions⁴⁶, and that steeper rivers might have been less prone to post-glacial colonisation. This habitat instability is higher for large values of range slope.”

Some interpretations still look quite a bit outstretched.. for example, it is not explained why "the region effect as well as the spatial autocorrelation factor appear as good surrogates of habitat complexity".

We removed that sentence which was not appropriate to explain the observed pattern. We now better explain the physical complexity hypothesis in the introduction (see above) and what we mean by habitat complexity (i.e. complex network of watercourses typically characterizing freshwater habitats vs. the more homogenous and continuous seawater habitat). We also added explanations for the effect of each predictor on freshwater fish diversity as well as their interpretation in the method section.

Also, it is important that the authors avoid sentences like "our results support the evolutionary speed hypothesis". There is no way to disentangle the "speed" and "colonization" hypotheses with this data. Therefore these results *do not* "support" either. They are consistent with both! This should be made absolutely clear to the readers.

We fully agree this is crucial point to avoid future misinterpretation by the readers.

We replaced « support » by « consistent with » in the discussion and we stated in the revised version that it remains impossible to disentangle some hypotheses using our data and approach.

Importantly, it is not explained how confidence intervals for model coefficients were obtained. As P-values are biased by model selection (Line 476), these CIs are the most important hint for the significance of predictors!

We now added explanations on the confidence interval in the legend of Figure 3.

« Confidence intervals were estimated from the standard error of each coefficient at a level of 5% and were obtained with the command confint in the R package lm. »

In addition, we removed the sentence on the biased values since we did not use these p-values to interpret the model. They just come from the selection procedure while the final models, from which we extracted the coefficients, were computed apart from the selection procedure.

I suggest some restructuring to the Discussion, which is quite redundant and not very linear (see the commented manuscript for details).

We restructured the discussion as recommended. See the new discussion lines 212-360.

Reviewer #4 (Remarks to the Author):

I believe that the authors revised the manuscript appropriately by taking all reviewers' considerations into account. Below, I provide a few minor comments to be addressed before publication of the manuscript.

We thank the reviewer for his positive comment.

While the authors lay out a clear expectation (the positive relationship between GD and SR) in lines 26-27, they do not mention that they are actually testing for that in the next lines (lines 29-30), but they mention it only as a result (lines 31-32).

We added a clearer sentence to address this concern lines 29-30: « we examined the correlation between genetic diversity and species diversity and further investigated the global distribution of intraspecific genetic diversity in relation to climate and geography”.

Please also mention in the methods the genetic marker (co1) that was used for the inferences

We added COI line 28.

Lines 144-147: The authors mention that a peak of genetic diversity occurs in the North latitudinal band of 10-20 degrees latitude (Fig. 1 b,d). However, I do not see this peak in the figure. I see peaks in the South Latitudinal bands (-10 to -20 degrees). Please correct.

Thank you. We corrected that point (lines 133-135).

Line 150: What do the authors mean by “median genetic diversity per cell”. I thought the genetic diversity was calculated as the mean across all sampled species per cell (395-396). Maybe they meant that they compared the median values of the two distributions (freshwater vs marine).

The reviewer is correct. Our sentence was confusing. We changed it: it now reads line 114 « we found that the median value of the genetic diversity per cell”

Line 179-178: Change “because detected” to “because it was identified as”

Changed (see line 162).

Line 192: I am missing a statement in the Methods on how the relative explained variance was calculated.

We added the explanation in the method, it reads “Relative variance of genetic diversity explained by the various factors was estimated and represented as partial plots with the package *hier.part*” lines 493-494”.

Discussion:

244-246: Is there a relevant reference that states that connectivity (i.e. gene flow) reduces genetic variation (i.e. number of mutations) at the nucleotide level? If so, please add it.

No, we are not aware of any reference.

Lines 318-324: I suggest to move this part in the previous paragraph (Line 299) where

the authors discuss the intra- interspecific diversity associations.

We agree with the reviewer. We considered this point when restructuring the discussion. The sentence is now in the previous paragraph and reads lines 295-303:

« However, the weakness of the relationship between genetic and species diversity also indicates that the processes underlying genetic diversity patterns might not be completely similar to those underlying species diversity patterns. These differences might be explained by disparities in temporal and spatial scales or in responses to environmental changes at which parallel ecological and evolutionary processes operate (mutation vs. speciation; genetic vs. ecological drift; gene flow vs. dispersal; selection vs. environmental filter)⁵⁴. »

Figures:

Figure 3: “Determinants” instead of “Determinant” in title?

Corrected

Supplementary Figure 4: Please add (a) and (b) letters in the subplots

We added a and b

Reviewers' Comments:

Reviewer #3:

Remarks to the Author:

I think the manuscript has further improved and is ready for publication. I am very glad that I could give my small contribution.

I have attached an edited version of the manuscript with a small number of further comments.